# Snow Water Equivalent Retrieval and Analysis Over Altay Using 12-Day Repeat-Pass Sentinel-1 Interferometry

Jingtian Zhou[1,2], Yang Lei[1], Jinmei Pan[1], Cunren Liang[3], Zhang Yunjun[2,4], Weiliang Li[1,2], Chuan Xiong[5], Jiancheng Shi[1], and Wei Ma[1]

[1]National Space Science Center, Chinese Academy of Sciences, Beijing, 100190, China
[2]University of Chinese Academy of Sciences, Beijing, 100049, China
[3]School of Earth and Space Sciences, Peking University, Beijing, 100871, China
[4]National Key Laboratory of Microwave Imaging, Aerospace Information Research Institute, Chinese Academy of Sciences, Beijing, 100094, China
[5]Southwest Jiaotong University, Faculty of Geosciences and Engineering, Chengdu, 611756, China

*Correspondence to*: Yang Lei (leiyang@nssc.ac.cn)

**Abstract.** Accurate Snow Water Equivalent (SWE) estimation is significant for understanding global climate change, surface energy balance, and regional water cycles. However, although many studies have examined the inversion of SWE using active and passive microwave remote sensing, it remains challenging to assess its global distribution with sufficient temporal and spatial resolution and accuracy. Interferometric Synthetic Aperture Radar (InSAR) has become a promising technique for SWE change estimation, which is limited by the optimal radar frequencies and revisit intervals that have not been available until recently. In this study, 12-day Sentinel-1 C-band InSAR data from 2019 to 2021 are used to retrieve ΔSWE (SWE changes in one InSAR pair) and cumulative SWE in the Altay region of Xinjiang, China. The correlation between the retrieved ΔSWE and in-situ observations reaches R = 0.56, with a low RMSE of 9.54 mm (n = 152) throughout the two whole snow seasons, with values of R = 0.58 and RMSE of 10.1 mm for 2019-2020, and R = 0.48 and RMSE of 8.6 mm for 2020-2021, respectively. These results are obtained by filtering wet snow. Heavy snowfall leads to decorrelation and phase unwrapping errors, which affect ΔSWE retrieval and are propagated into cumulative SWE. Validation of the cumulative SWE after removing wet snow yields an RMSE of 40.9 mm, which improves to 28.3 mm when high-elevation stations with unwrapping errors due to heavy snowfall are also excluded. InSAR-derived cumulative SWE time series show consistency with ground observations at some stations, though slight overestimations and underestimations are observed due to error accumulation. Various factors combined with validation results show that higher coherence, lower air temperature, and reliable snow density improve the retrieval accuracy. The proposed coherence-weighted least squares phase calibration method demonstrates that selecting at least half of the available in-situ ΔSWE stations for calibration yields reliable ΔSWE estimates, although including more points can further improve the robustness. Calibrating only the integer multiples of $2\pi$ provides reasonable accuracy, but is still inferior to the full calibration method, indicating that residual modulo $2\pi$ phase has a noticeable contribution to the final inversion accuracy, which highlights that phase calibration plays a key role in the accurate ΔSWE retrieval. This study provides a valuable reference and processing prototype for applying 12-day revisit Sentinel-1 and future NISAR InSAR data to SWE monitoring.

## 1 Introduction

Snow significantly influences the balance of surface radiation energy due to its high albedo, thermal insulation properties, and heat absorption during melting periods (You et al., 2020). These characteristics make snow an essential indicator of the global climate system (Aguirre et al., 2018). Snowmelt is a crucial source of water resources to billions of people worldwide (Barnett et al., 2005). Snow water equivalent (SWE) is defined as the height of liquid water that would be produced if a snow column of a specified thickness completely melts into water. SWE is a crucial input parameter in hydrological processes, ecological models, and climate system models (Derksen et al., 2010). It also plays a key role in the energy transfer process between soil and atmosphere. However, evaluating the global distribution of SWE with adequate temporal and spatial resolution and accuracy remains challenging.

Passive microwave (PM) remote sensing, based on the microwave emissions from snowpack (Foster et al., 1997), is currently the main method of retrieving daily spatiotemporal information on SWE at a large scale. This method will become saturated for SWE larger than 150 mm, which limits their use in mountainous areas. Many research has been conducted using passive microwave remote sensing to estimate snow depth and SWE (Takala et al., 2011; Dai Liyun et al., 2012; Tedesco and Jeyaratnam, 2016). While satellite-based passive microwave remote sensors have provided valuable insights for global estimation of cryosphere snow depth (SD)/SWE, they have limited spatial resolution, typically at the 10-kilometer level. Although a large amount of efforts have provided accurate SWE products using PM observations, existing SWE products still do not meet the minimum accuracy requirements for hydrological applications (Brown et al., 2018).

Active microwave (radar) has shown stronger applicability in basin-scale snow research due to its high spatial resolution (tens of meters typically) and sensitivity to snow parameters (Storvold et al., 2006; Shi and Dozier, 1996; Thakur et al., 2012). This technique relies on backscattering from the volume scattering of snow. Higher frequencies (Ku and X-band) have been used to estimate SWE (Rott et al., 2010; Yueh et al., 2009; King et al., 2018; Zhu et al., 2021). However, a single parameter retrieval of SWE is challenging. This is because radar backscatter depends on multiple factors, such as snow density, snow depth, liquid water content, stratigraphy, grain size, and soil/vegetation conditions, as well as systematic factors (frequency and polarisation). Moreover, snow microstructure parameters are hard to assess over a large scale (Rutter et al., 2019).

Recently, repeat-pass Interferometric Synthetic Aperture Radar (InSAR) offers a promising approach to obtaining SWE changes at high spatial resolution and accuracy (e.g., 15 mm at L-band, 3.75mm at C-band.) by capturing radar phase changes. The method for retrieving SWE using InSAR was first proposed by Guneriussen (2002). A key advantage of this approach is that low-frequency signals are hardly affected by snow stratigraphy, and knowledge of snow microstructure is not required (Yueh et al., 2017). Subsequently, the technique was applied under various conditions, including a range of frequencies, temporal baseline pairs, and different acquisition platforms. It was applied to C-band spaceborne repeat-pass InSAR datasets from ERS with a short temporal baseline of 3-day which was conducted on the Austrian Alps (Rott et al., 2003) and the North Slope of Alaska (Deeb et al., 2011). The C-band spaceborne repeat-pass InSAR datasets from Sentinel-1 with a 6-day revisit during winter over Idaho are applied to retrieve SWE (Oveisgharan et al., 2024). The higher frequency X/Ku-band was

explored using dense time series from a ground-based radar (Leinss et al., 2015). Demonstrations of low-frequency L-band were based on a variety of airborne InSAR data, such as a 4-month dataset from DLR's E-SAR (Rott et al., 2003), 12-day pairs from NASA/JPL's UAVSAR (Marshall, 2020), and 8-day temporal baselines also from UAVSAR (Hoppinen et al., 2023). Additionally, temporal baselines ranging from 5 to 20 days were analyzed from UAVSAR pairs in forested areas (Bonnell et al., 2024). Spaceborne L-band, 4-month InSAR pairs from ALOS-2 were examined over regions with sparse vegetation (Lei et al., 2023). In the Altay region of Xinjiang province, China, available historical L/C-band InSAR datasets (e.g., JAXA's ALOS, ESA's Sentinel-1, and China's Lutan-1) were utilized to produce SWE change products (Lei et al., 2024). These investigations demonstrate that low-frequency radar signals, combined with shorter revisit times, can enhance penetration and reduce temporal decorrelation. This makes them particularly suitable for monitoring SWE in areas with frequent snowfall. Nevertheless, the limited availability of satellite observations with suitable frequencies and temporal baselines causes a challenge to the widespread application of this technique.

At present, Sentinel-1 data with a 6-day revisit period and InSAR method have been used to retrieve SWE in Idaho, USA, and good results have been obtained (Oveisgharan et al., 2024). However, the use of spaceborne data and the InSAR method for SWE retrieval has not been widely examined. In most regions globally, only a 12-day revisit period of Sentinel-1 data can be achieved (Kellndorfer et al., 2022). The retrieval performance under a 12-day revisit period with C-band spaceborne data has not been well studied. In this study, we evaluated the performance of SWE retrieval over Altay using interferometry based on 12-day C-band Sentinel-1 data. In Sect. 2, we introduce the study area and dataset used. Sect. 3 describes the methodology we use, which shows how we processed Sentinel-1 data and retrieved it to SWE. Sect. 4 introduces the comparison between the retrieved SWE with in-situ data, followed by factors that may influence the results in Sect. 5. At last, the conclusions are provided in Sect. 6.

## 2 Study Area and Datasets

### 2.1 Study Area

Altay Prefecture (44°59′35″–49°10′45″N, 85°31′57″–91°01′15″E) of Xinjiang province is situated in northwestern China, covering a total area of approximately 118,000 km$^2$, bordering Kazakhstan, Russia, and Mongolia. Altay Prefecture is one of the regions with rich seasonal snowmelt water resources, providing snow water resources for these four countries. The average annual snow depth is approximately 40 cm, with a maximum of over 70 cm (Dai et al., 2022). The period of snow accumulation and ablation typically occurs from October to late April or early May, spanning roughly 6–7 months. The snow density is small, with a typical value of 0.2 g·cm$^{-3}$ (Yue et al., 2017). The region has a temperate continental climate, featuring short, warm summers and long, cold, snowy winters, with a mean annual temperature of 0.7–4.9 °C (Fu et al., 2017). The terrain is low in the southwest and high in the northeast (Fig. 1). In Xinjiang, China, the Altay region exhibits varied topography, with elevations exceeding 3000 m in the northeast, 700–800 m in the central part, and about 600 m in the southwest. The core study area spans 47.5–48.5°N and 87.5–89.5°E, covering roughly 110 × 150 km.

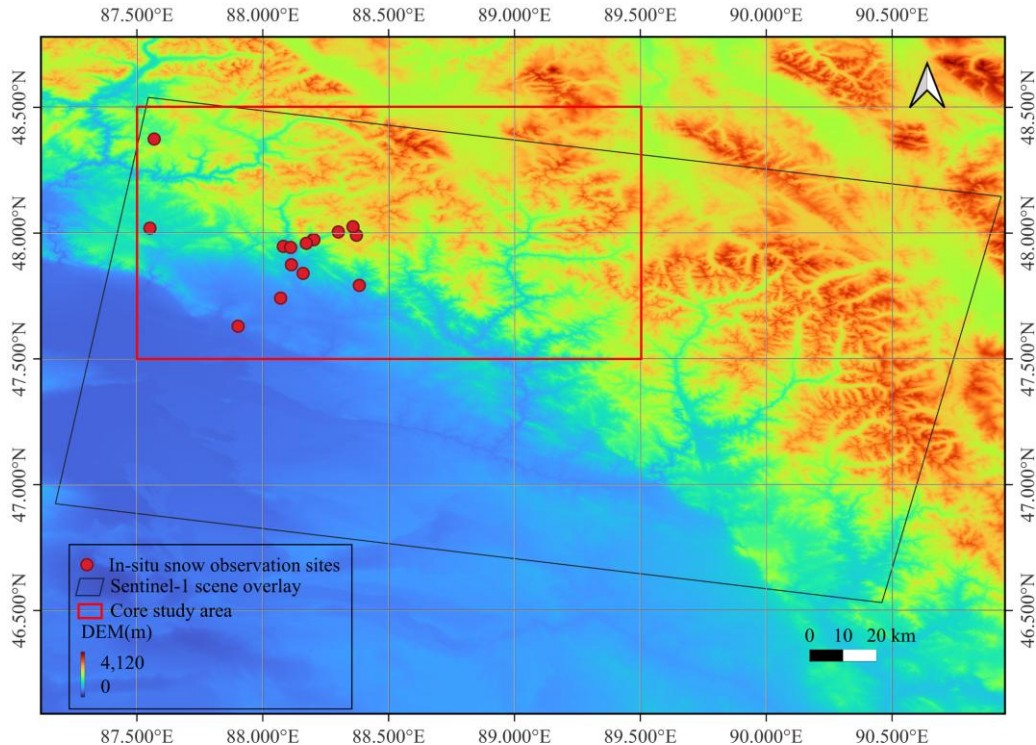

**Figure 1: Location of study area, including 15 in-situ snow station points.**

 **2.2 Datasets**

**2.2.1 Sentinel-1**

The European Space Agency's (ESA) Copernicus Sentinel-1 mission began in 2014 with the launch of Sentinel-1A on 3 April 2014, followed by Sentinel-1B on 25 April 2016. Each satellite has a 12-day repeat cycle. They orbit 180° apart, together imaging the Earth every 6 days but only in limited regions, which are predominantly over Europe (Kellndorfer et al., 2022). Sentinel-1 supports dual polarization and delivers products quickly. The data can be freely accessed from the Alaska SAR Facility (ASF, https://search.asf.alaska.edu/). The Sentinel-1 radar operates at C-band (5.405GHz) and offers four imaging modes. These modes vary in resolution, reaching as fine as 5 m, and cover up to 400 km. The operational mode used in this study is the Interferometric Wide swath (IW) mode, which operates as TOPS mode, offering a large swath width of 250 km with a ground resolution of 5×20 m in range and azimuth, respectively (Torres et al., 2012). Hence, a 15×5 (range×azimuth) multilooking is applied, resulting in a final resolution of 75×100 m. For this study, Sentinel-1 Single Look Complex (SLC) data is collected over the Altay region. 19 scenes were acquired every 12 days from September 5, 2019, to April 8, 2020, and 18 scenes from September 11, 2020, to April 3, 2021. All data correspond to a descending flight direction with an overpass time of approximately 00:13 UTC, with the local Beijing time being 08:13 (UTC+8), path 19, frame 434.

### 2.2.2 In-situ snow observations

The observation data of snow parameters, including snow depth and SWE, is collected from in-situ sites established by the Altay Meteorological Institute and from our own established observation stations. A total of 15 sites are available from 2019 to 2021. These sites are primarily situated in flat areas to minimize the influence of surrounding vegetation. Among these sites, only two measure SWE using snow pillow, while the remaining 13 sites measure snow depth. Snow depth sites use lasers, or a combination of snow poles and time-lapse cameras. The snow depth obtained by laser is automatically obtained with a shorter

interval of 10 minutes or one hour. However, the snow depth of the photographic snow observation station needs to be read manually with a slightly longer interval of 3–4 hours. The locations and environments of the snow depth measurement sites using snow poles and cameras are shown in Fig. 2. SWE data are collected less frequently (every 3–7 days). For validation, snow depth observations nearest to satellite overpass times are converted to SWE using snow density from ERA5-Land hourly data.

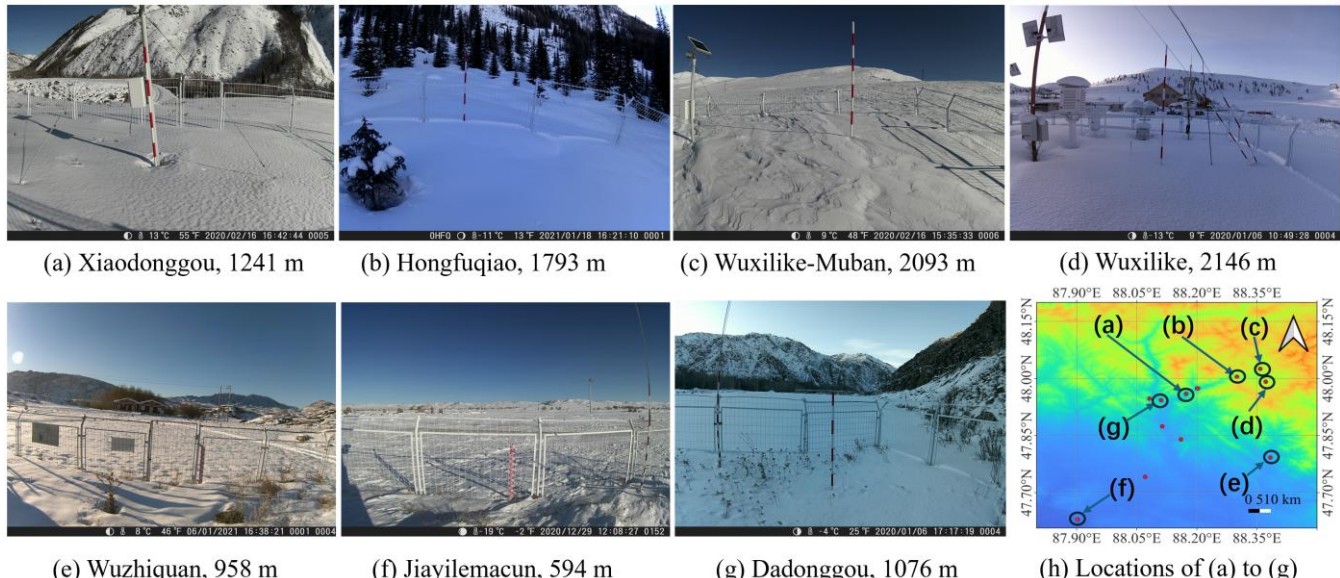

**Figure 2: In-situ snow depth observation sites using depth poles captured by the camera. There are several typical kinds of land cover, including valley sites: (a) Xiaodong and (g) Dachonggou, conifer forest in the shade aspect hill: (b)**

**Hongfuqiao, plain sites:(c) Wuxilike-Muban, (d) Wuxilike, (g) Jiayilemacun and (h) Wuzhiqhuan. (h) shows these sites' locations in the study area.**

### 2.2.3 Auxiliary data

In this study, the auxiliary data are acquired for the following purposes: snow density is utilized to compute SWE for multiple snow depth (SD) stations, air temperature data support subsequent analysis, and digital elevation models (DEMs) along with precise orbit data are necessary for the InSAR processing of Sentinel-1 SLC data.

ERA5-Land hourly data is a reanalysis dataset (https://cds.climate.copernicus.eu/datasets/reanalysis-era5-land?tab=overview) providing a consistent view of the evolution of land variables over several decades, with enhanced spatial resolution compared to ERA5. The data are freely available through Climate Data Store. In this study, we obtain ERA5 air temperature and snow density data for our study area at 00:00 UTC, which is the closest available time to the Sentinel-1 overpass at 00:13 UTC. Air temperature is defined at 2 m above the surface and is calculated by interpolating between the lowest model level and the Earth's surface, taking atmospheric conditions into account. Temperature values in Kelvin can be converted to degrees Celsius (°C) by subtracting 273.15.

Snow density is derived from the ECMWF Integrated Forecast System (IFS) model, which represents snow as a single additional layer over the uppermost soil level. Notably, ERA5-Land snow density has a horizontal resolution of $0.1° × 0.1°$ (~9 km), which is much coarser than the spatial resolution of Sentinel-1 InSAR data (tens of meters) used for SWE inversion. This spatial scale mismatch implies that ERA5-Land snow density represents average conditions over large areas and may not capture the fine-scale variability captured by InSAR. Therefore, in addition to potential uncertainties in the ERA5-Land snow density data itself, this mismatch can introduce errors when using ERA5 snow density for in-situ SWE calculation.

For Sentinel-1 SLCs' InSAR process input data, digital elevation models (DEMs) and precise orbit data are acquired from Shuttle Radar Topography Mission (SRTM) DEM and the ASF (https://s1qc.asf.alaska.edu/aux_poeorb/), respectively.

### 3 Methodology

To achieve the goal of assessing the performance of 12-day Sentinel-1 C-band InSAR for monitoring SWE changes in the whole dry snow season, a series of components is described, including the theory of SWE retrieval from interferometric phase, data processing procedures, and the phase calibration method. In particular, Sect. 3.1 explains the theory of InSAR-derived SWE. Sect. 3.2 describes the workflow of stack processing Sentinel-1 SLCs to generate the interferometric phase of nearest neighbour dates, then processing of InSAR phases to produce a time series phase changes after correction of atmospheric delay and DEM error, and finally converting the phase change to SWE change. Sect. 3.3 introduces the in-situ SWE processing method. Sect. 3.4 provides the phase calibration method for InSAR-derived SWE change by using in-situ measurements for calibration.

### 3.1 Relationship between $\Delta SWE$ and $\Delta\phi$

The InSAR SWE retrieval method considers the signal penetration through the snow layer to the ground. The primary backscattering contribution from dry snow-covered terrain originates from the snow–ground interface, while the volume

scattering effect on the interferometric phase is negligible, as confirmed by ground-based experiments (Matzler, 1996). The complex permittivity properties of snow, which are strongly dependent on the liquid water content, govern the propagation of radar waves in snow. At C-band, dry snow has a typical penetration depth of 20 m (Matzler, 1996; Rott et al., 2003), while wet snow with liquid water content is limited to a few centimeters due to a prominent rise in imaginary part of permittivity as water content increases.

The real parts of the complex permittivity $\varepsilon_s$ is a function of snow density as shown in Eq. (1) (Matzler, 1996):

$$\varepsilon_s = 1 + 1.60\rho_s + 1.86\rho_s{}^3 \tag{1}$$

where $\rho_s$ is expressed in g/cm³.

Because snow has a different dielectric constant from air, radar waves undergo refraction as they propagate through a snow layer. When comparing the optical path lengths of radar waves without and with snow conditions, a path delay can be observed. The delay arises from the change in optical path length, given by $n \cdot s$ (where $n$ is the refractive index and $s$ is the geometric path length), caused by refraction within the snowpack and the reduced propagation velocity of radar waves in snow compared to air. The signal delay can be derived from the geometry path illustrated in Fig. 3. For conditions without snow, the radar wave travels the distance CA, while for snow-covered conditions, the distance is DE+EA. Furthermore, this path delay also occurs when there is a change in snow depth $\Delta Z_s$, between two measurements, with the delay being proportional to $\Delta Z_s$. This delay in path length induces a differential interferometric synthetic aperture radar (DInSAR) phase difference, which can be correlated with the change in snow depth.

The relationship between change in SWE ($\Delta SWE$) and the differential interferometric phase shift ($\Delta\phi$), following Guneriussen et al. (2002), can be expressed based on the geometric configuration in Fig. 3 as :

$$\Delta\phi = 2k_i \cdot \Delta Z_s \left( \cos\theta - \sqrt{\varepsilon_s - sin^2\theta} \right) \tag{2}$$

Leinss et al. (2015) derive a nearly linear dependence between $\Delta SWE$ and $\Delta\phi$ by approximating the snow density dependent permittivity term from Eq. (1) into Eq. (2) and using a Taylor expansion under low density and small incidence angle assumptions, leading to the simplified expression in Eq. (3):

$$\Delta\phi = 2k_i \cdot \frac{\alpha}{2}\left(1.59 + \theta^{\frac{5}{2}}\right) \cdot \Delta SWE \tag{3}$$

where $\Delta\phi$ is the interferometric phase, $\Delta SWE$ is the SWE change, and the wavenumber is defined by $k_i = \frac{2\pi}{\lambda}$ with $\lambda$ being the central wavelength of the radar. The incidence angle at the air-snow interface is given by $\theta$. The optimal $\alpha$ is close to 1 for common incidence angles ($< 50°$). We utilize Eq. (3) to retrieve SWE, with the parameter α set to 1 in this study.

The main advantage of this method is its simplicity and a reduced reliance on a priori information. However, its application is constrained by several factors: (1) temporal decorrelation (Zebker and Villasenor, 1992; Jung et al., 2016; Lee et al., 2013), which is particularly critical for C-band data with a 12-day revisit cycle and can be severe over vegetated terrain; (2) the phase unwrapping problem (Engen et al., 2003; Rott et al., 2003; Leinss et al., 2015), which occurs when the SWE change is about

larger 30 mm in our cases ($2\pi$ phase change corresponds to 30 mm SWE changes); and (3) signal attenuation in wet snow, which limits the method to dry snow conditions (Storvold et al., 2006).

To improve the reliability of the inversion results, data corresponding to wet snow conditions were filtered out during the validation process on a per-site basis. Wet snow at the current observation site is identified based on any of the following criteria:

(a)    air temperature is above 0 °C at the current site for any acquisition date of the interferometric pair;

(b)    after February 1, if the coherence between two consecutive interferometric pairs drops by more than 0.3 at the current site, the second interferometric pair and all subsequent points for that site are excluded;

(c)    the coherence of the interferometric pair is below 0.35 at the current site.

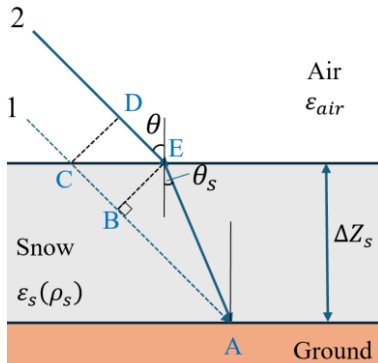

**Figure 3: Propagation path of radar wave through atmosphere with snow-free and snow-covered ground for a fixed pixel. $\theta$ is the incidence angle, $\theta_s$ is the refracted angle in snow, $\varepsilon_s$ is the real part of the permittivity of the snow, $\varepsilon_{air}$ is the real part of the permittivity of the air, and $\Delta Z_s$ is the increase in snow depth between the snow-free and snow-covered ground images.**

## 3.2 Sentinel-1 interferometric phase processing methods and procedures

The InSAR stack processing is performed using the NASA/JPL's open-source software ISCE2 (https://github.com/isce-framework/isce2) along with the time series tool MintPy (https://github.com/insarlab/MintPy). As shown in Fig. 4, the workflow consists of three main blocks: (i) InSAR stack processing for Sentinel-1 TOPS data using ISCE (Fattahi et al., 2016), (ii) InSAR time series analysis from a stack of unwrapped interferograms to phase time-series using MintPy (Yunjun et al., 2019), and (iii) phase calibration and SWE inversion.

In the first stage, after the co-registration, filtering, and phase unwrapping procedures, stacks of all secondary single-look complex (SLC) images are co-registered to the reference SLC. A coregistered stack of SLCs are produced, and the burst interferograms are merged. Merged interferograms are multilooked, filtered and unwrapped. A multi-look averaging of 15×5

(range×azimuth, similar to the following) is applied to 37 SLC data scenes, resulting in a ground resolution of 75×100 m. The SNAPHU algorithm is chosen for phase unwrapping in ISCE2.

In the second stage, the outputs from the first stage are processed to generate a corrected phase time series, which is then geocoded. Errors in phase unwrapping, tropospheric delays, and topographic residuals are corrected. The tropospheric delay correction uses the PyAPS method (https://github.com/insarlab/PyAPS), which estimates differential phase delay maps based on ECMWF's ERA-5 data. To prevent the removal of long-term trends that may impact SWE inversion, the deramp step in MintPy is not used in our study.

In the third stage, the phase time series is calibrated using in-situ ΔSWE measurements. After calibration, the corrected phase measurements are used to derive ΔSWE. The details of the phase calibration method are provided in Sect. 3.4.

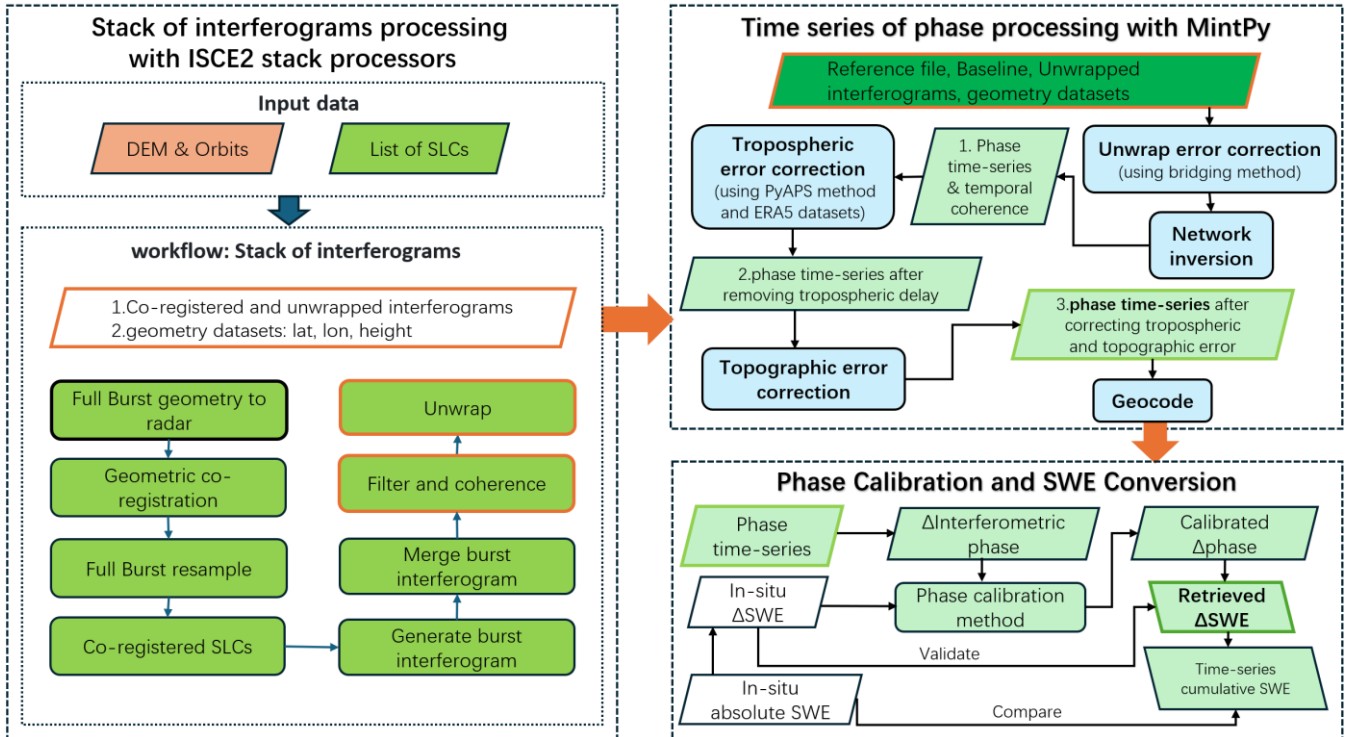

**Figure 4: Flow chart of the time series InSAR Processing procedures.**

### 3.3 In-situ SWE processing method

In-situ snow ground observations consist of measurements of snow depth and SWE.

To ensure spatiotemporal consistency with satellite overpasses, snow depth data recorded closest to the satellite observation time are selected (00:00 UTC, 08:00 Beijing local time). When minor gaps exist in the snow depth time series on the required

dates, missing values are filled by averaging observations from adjacent available dates. Subsequently, snow density from the
ERA5-Land dataset corresponding to the same location and time is extracted. SWE is then estimated by multiplying the snow
depth by the corresponding ERA5-Land snow density.

For SWE data obtained from snow pillow observations, which are generally reliable but recorded every few days, daily
interpolation is required. A direct average from adjacent days cannot be applied due to the relatively long observation intervals.
Therefore, daily ERA5-Land SWE data closest to the satellite observation time are used as a reference. A least squares fitting
method is applied to determine a scaling factor that brings the ERA5-Land SWE values closer to the in-situ observations. The
scaled ERA5 SWE data are then used to interpolate the in-situ SWE observations through a fifth-order polynomial fitting. This
approach enables the construction of a continuous daily SWE time series, ensuring a daily dataset aligned with the satellite's
12-day revisit cycle.

### 3.4 Phase calibration based on in-situ SWE

InSAR phase measurements are relative and must be calibrated to remove the unknown scene-wide phase offset. Several
factors contributing to phase offset: integer multiple of $2\pi$, data processing (focusing, range gating), DEM residual error,
unwrapping error, atmospheric (troposphere and ionosphere) phase delay, systematic phase calibration error. In geographical
applications of InSAR, the reference point used for calibration is typically chosen such that the displacement remains
unchanged or is known between the two image acquisitions. For ΔSWE retrieval using InSAR, previous studies have suggested
two strategies: using corner reflectors with snow always being cleaned, which offer a stable zero-phase reference point (Nagler
et al., 2022; Dagurov et al., 2020) or using the average of in situ ΔSWE measurements (Conde et al., 2019; (Oveisgharan et
al., 2024).

In our case, it is difficult to identify a fixed phase reference point because snow accumulation and ablation affect the entire
scene, and corner reflector deployment is labor-intensive. Therefore, we follow the second strategy and use all available in situ
stations for calibration rather than a subset. Using a larger number of spatially distributed stations reduces random errors and
minimizes potential biases introduced by individual stations. This approach ensures a more stable and unbiased calibration of
the InSAR-derived $\Delta SWE$. At the same time, our method differs from previous studies in one aspect: instead of directly
calibrating the retrieved ΔSWE using in-situ observations, we use all available in situ SWE measurements to calibrate the
interferometric phase itself. The calibrated phase is then used in the $\Delta SWE$ retrieval. If local incidence angles are taken into
account, a scene-wide phase offset produces different $\Delta SWE$ offsets at each site. Therefore, applying a single $\Delta SWE$
calibration to the entire scene is not suitable, and calibrating the phase directly is more appropriate, reducing incidence-angle–
related uncertainties and improving the robustness of the $\Delta SWE$ estimates.

### 3.4.1 Phase calibration method

Based on this strategy, we formulate the calibration of the interferometric phase using in situ SWE observations. For a single
InSAR scene, the phase calibration equations can be expressed from Eq. (3) as follows:

$$\Delta\phi - C = 2k_i \cdot \frac{\alpha}{2}\left(1.59 + \theta^{\frac{5}{2}}\right) \cdot \Delta SWE \tag{4}$$

where $C$ is the phase calibration constant for the interferogram, which includes both the integer multiple ambiguity of $2\pi$ and residual phase as mentioned in Sect. 3.4. Let $y = 2k_i \cdot \frac{\alpha}{2}\left(1.59 + \theta^{\frac{5}{2}}\right) \cdot \Delta SWE$, where $y$ represents the calculated interferometric phase from in-situ $\Delta SWE$ measurements. In the following, we present the application of our phase calibration method in three representative cases: (A) a single InSAR scene, (B) multiple non-redundant InSAR scenes, and (C) redundant InSAR scenes (simple example).

**(A) A Single InSAR Scene**

Suppose there are $m$ in-situ SWE observations at different locations within a single InSAR scene. This can be formulated as:

$$y = \Delta\phi + AC \tag{5}$$

where $y = (y_1 \quad y_2 \quad \cdots \quad y_m)^T$ is an $m \times 1$ vector of calculated interferometric phases from in-situ data, with each element $y_i$ corresponding to the $i$-th in-situ SWE measurement; $\Delta\phi = (\Delta\phi_1 \quad \Delta\phi_2 \quad \cdots \quad \Delta\phi_m)^T$ is an $m \times 1$ vector of measured interferometric phases from this InSAR pair, with $\Delta\phi_i$ denoting the observed interferometric phase at the location of the $i$-th in-situ measurement; $A = (-1 \quad -1 \quad \dots \quad -1)^T$ is an $m \times 1$ design matrix

To estimate the phase calibration constant $C$, we employ a coherence-weighted least squares approach. Then the optimal coherence-weighted least squares coeficients for this single InAR scene is

$$\hat{C} = (A^T W A)^{-1} A^T W (y - \Delta\phi) = \frac{\sum_{i=1}^{m} \gamma_i (\Delta\phi_i - y_i)}{\sum_{i=1}^{m_n} \gamma_i} \tag{6}$$

where $W = diag(\gamma_1, \gamma_2, \dots, \gamma_m)$ is an $m \times m$ diagonal weight matrix based on coherence values $\gamma_i$, with $\gamma_i$ representing the coherence value at the $i$-th measurement location. This formulation assigns a larger weight to phases derived from locations with higher coherence, enhancing the robustness of the calibration.

**(B) Multiple Non-redundant InSAR Scenes**

We now extend the phase calibraton method to the case of $N$ ($N > 1$) non-redundant interferometric pairs. Each pair requires estimation of a distinct phase calibration constant $C_n$. For the $n$-th ($n$=1, 2, …, $N$) interferometric pair, let $m_n$ denote the number of in-situ SWE observations, which may vary from pair to pair. Let $\Delta\phi_{m_n}$ and $y_{m_n}$ represent the $m_n \times 1$ vector of observed interferometric phases and the corresponding in-situ calculated interferometric phases for the $n$-th pair, respectively. Bold represents vector (lower case) and matrix (capitalization).

The equations for all pairs can be expressed in matrix form identical to that of Eq. (5): where $y = (y_{m_1} \quad y_{m_2} \quad \cdots \quad y_{m_N})^T$ is an $m \times 1$ vector of stacked calculated interferometric phases, with $m = \sum_{i=1}^{N} m_i$; $\Delta\phi = (\Delta\phi_{m_1} \quad \Delta\phi_{m_2} \quad \cdots \quad \Delta\phi_{m_N})^T_{m \times 1}$ is an $m \times 1$ vector of stacked measured interferometric phases; $C = (C_1 \quad C_2 \quad \dots \quad C_N)^T$ is an $N \times 1$ vector of calibration constants; $A = blkdiag(v_{m_1}, v_{m_2}, \dots, v_{m_N})$ is an $m \times N$ block-diagonal design matrix, where each block $v_{m_n} = (-1 \quad -1 \quad \dots \quad -1)^T$ is an $m_n \times 1$ vector.

$$A = \begin{pmatrix} v_{m_1} & 0 & \cdots & \cdots & 0 \\ 0 & v_{m_2} & 0 & \cdots & \vdots \\ \vdots & 0 & v_{m_3} & \cdots & \vdots \\ \vdots & \vdots & \vdots & \ddots & 0 \\ 0 & \cdots & \cdots & 0 & v_{m_N} \end{pmatrix}_{m \times N} \tag{7}$$

The coherence-weighted least squares estimate of $C$ is given by:

$$\widehat{C} = (A^T W A)^{-1} A^T W (y - \Delta\phi) \tag{8}$$

Where $W$ is an $m \times m$ block-diagonal weight matrix constructed from the coherence values $\gamma$ of all measurement points:

$$W = blkdiag(w_{m_1}, w_{m_2}, \ldots, w_{m_N}) = \begin{pmatrix} w_{m_1} & 0 & \cdots & \cdots & 0 \\ 0 & w_{m_2} & 0 & \cdots & \vdots \\ \vdots & 0 & w_{m_3} & \cdots & \vdots \\ \vdots & \vdots & \vdots & \ddots & 0 \\ 0 & \cdots & \cdots & 0 & w_{m_N} \end{pmatrix}_{m \times m} \tag{9}$$

And each diagonal block $w_{m_n}$ is an $m_n \times m_n$ matrix:

$$w_{m_n} = diag(\gamma_1, \gamma_2, \ldots, \gamma_{m_n}) = \begin{pmatrix} \gamma_1 & 0 & \cdots & \cdots & 0 \\ 0 & \gamma_2 & 0 & \cdots & \vdots \\ \vdots & 0 & \gamma_3 & \cdots & \vdots \\ \vdots & \vdots & \vdots & \ddots & 0 \\ 0 & \cdots & \cdots & 0 & \gamma_{m_n} \end{pmatrix}_{m_n \times m_n} \tag{10}$$

Each interferometric phase image is calibrated based on its estimated $\widehat{C_n}$ ($n = 1,2,3 \ldots, N$). The coherence-weighted least squares method provides an unbiased estimate of $C$. Specially, when only nearest without redundant InSAR pairs are considered, $A$ is a block-diagonal matrix with $v_{m_n}$ blocks that are being independent; thus, the solutions to each $\widehat{C_n}$ can be derived separately. In this scenario, for the $n$-th InSAR pair, the estimated $\widehat{C_n}$ represents the coherence-weighted average value of all calibration parameters at different locations:

$$\widehat{C_n} = \left(v_{m_n}{}^T w_{m_n} v_{m_n}\right)^{-1} v_{m_n}{}^T w_{m_n} x_{m_n} = \frac{\sum_{i=1}^{m_n} \gamma_i (\Delta\phi_i - y_i)}{\sum_{i=1}^{m_n} \gamma_i} \tag{11}$$

Note that up to this equation, the coherence-weighted least-squares solution is strictly equivalent to calibrating each interferogram individually, and the matrix formulation does not provide additional benefit unless redundant interferograms are involved.

In this study, only the nearest InSAR pair is considered because the temporal baseline in our study is already 12 days, which is relatively long compared to the rate of snow variation. Therefore, we adopt the non-redundant multi-pair approach described in this method. Using longer baselines, such as 24 or 48 days, is less advantageous for phase unwrapping. However, if shorter temporal baseline data become available in the future, redundant interferometric pairs should be considered. In that scenario, the design matrix $A$ is no longer block-diagonal, and the coherence-weighted least squares solution from Eq. (8) instead of Eq. (11) must be applied.

## (C) Redundant InSAR Scenes: A Simple Example

To illustrate the case of redundant interferometric pairs, consider a sequence of $n$ SAR acquisitions. Then, $n - 1$ adjacent interferometric pairs can be formed. The number of all possible interferometric pairs is $\binom{2}{n} = \frac{n(n-1)}{2}$. For simplicity, we examine a three-scene case (1st, 2nd and 3rd), which yields three combinations of interferometric pairs. The calibration constants satisfy the relation theoretically $C_{13}=C_{12}+C_{23}$, where $C_{ij}$ denotes the calibration parameter for the interferogram formed from the $i$-th and $j$-th SAR acquisitions.

In this case, let $\Delta\boldsymbol{\phi}_{ij}$ and $\boldsymbol{y}_{ij}$ be the $m_{ij} \times 1$ vectors of the observed interferometric phases and the phases calculated from in-situ SWE measurements, respectively, for the pair $(i, j)$. Note that $m_{ij}$, the number of in-situ SWE observations, may differ for each pair. The phase calibration can be also expressed in the same form as Eq.(5) with the following definitions: $M = m_{12} + m_{23} + m_{13}$ is the total number of observations; $\boldsymbol{y} = (\boldsymbol{y_{12}} \quad \boldsymbol{y_{23}} \quad \boldsymbol{y_{13}})^T{}_{M\times 1}$ is the stacked vector of calculated phases; $A = blkdiag(\boldsymbol{v_{12}}, \boldsymbol{v_{23}}, \dots, \boldsymbol{v_{13}})$ is an $M \times 3$ block-diagonal design matrix, where each block $\boldsymbol{v_{ij}} = (-1 \quad -1 \quad \dots \quad -1)^T$ is an $m_{ij} \times 1$ vector; $\boldsymbol{C} = (C_{12} \quad C_{23} \quad C_{13})^T{}_{3\times 1}$ is the vector of calibration constants; $\Delta\boldsymbol{\phi} = (\Delta\boldsymbol{\phi}_{12} \quad \Delta\boldsymbol{\phi}_{13} \quad \Delta\boldsymbol{\phi}_{13})^T{}_{M\times 1}$ is the stacked vector of observed interferomtric phases.

Substituting those definitions into Eq.(13) yields the matrix form:

$$\begin{pmatrix} \boldsymbol{y_{12}} \\ \boldsymbol{y_{23}} \\ \boldsymbol{y_{13}} \end{pmatrix}_{M\times 1} = \begin{pmatrix} \boldsymbol{v_{12}} & 0 & 0 \\ 0 & \boldsymbol{v_{23}} & 0 \\ 0 & 0 & \boldsymbol{v_{13}} \end{pmatrix}_{M\times 3} \begin{pmatrix} C_{12} \\ C_{23} \\ C_{13} \end{pmatrix}_{3\times 1} + \begin{pmatrix} \Delta\boldsymbol{\phi}_{12} \\ \Delta\boldsymbol{\phi}_{23} \\ \Delta\boldsymbol{\phi}_{13} \end{pmatrix}_{M\times 1} \tag{12}$$

Since C is a systematic calibration parameter, it follows that $C_{13} = C_{12} + C_{23}$. Substituting this constraint into Eq.(12) can be rewritten as:

$$\begin{pmatrix} \boldsymbol{y_{12}} \\ \boldsymbol{y_{23}} \\ \boldsymbol{y_{13}} \end{pmatrix}_{M\times 1} = \begin{pmatrix} \boldsymbol{v_{12}} & 0 \\ 0 & \boldsymbol{v_{23}} \\ \boldsymbol{v_{12}} & \boldsymbol{v_{23}} \end{pmatrix}_{M\times 2} \begin{pmatrix} C_{12} \\ C_{23} \end{pmatrix}_{2\times 1} + \begin{pmatrix} \Delta\boldsymbol{\phi}_{12} \\ \Delta\boldsymbol{\phi}_{23} \\ \Delta\boldsymbol{\phi}_{13} \end{pmatrix}_{M\times 1} \tag{13}$$

By incorporating the redundant interferometric pair $C_{13}$ into the design matrix A, the number of equations for solving $C$ increases, and the matrix becomes non-block-diagonal with non-zero elements introduced in its off-diagonal regions. As a result, the dimension of the parameter vector $\boldsymbol{C}$ is reduced, while the forms of $\boldsymbol{y}$ and $\Delta\boldsymbol{\phi}$ remain unchanged. This demonstrates how redundancy can be exploited to improve the estimation of calibration parameters in the presence of multiple interferometric pairs.

### 3.4.2 Phase calibration using different numbers of randomly selected in-situ stations

In Sect. 3.4.1, the calibration parameter for each interferometric pair is estimated by coherence-weighted least squares, where all available in situ stations are used. In this section, to validate the phase calibration method and to assess its performance under different conditions, we test the calibration using different numbers of in-situ stations.

For each interferometer pair, only a subset of available in-situ $\Delta SWE$ observations is used to derive the calibration parameter (calibration points), while the remaining stations are reserved for validation only. Because the number of available stations is

relatively limited after removing missing observations or filtering wet snow (in our case, the maximum is 12), we do not apply additional selection criteria (e.g., air temperature, elevation). In particular, both the lowland (valley/plain) stations and the

high-elevation mountain stations are randomly selected for calibration and validation, without any preference. Instead, a Monte Carlo random selection strategy is adopted: from all available stations, a specified number (1, 2, …, up to 9) is randomly chosen for calibration, and the remaining stations (at least three) are used for validation. This random selection is repeated 100 times for each case, and the statistics of these realizations are used to quantify the performance. This procedure ensures a fair evaluation of how the number of stations used for calibration affects the robustness of the phase calibration and subsequent

$\Delta SWE$ retrieval.

We note that, except for this specific analysis (Sect. 3.4.2) and its corresponding validation results (Sect. 4.3.2), all other retrieval results presented in this study are based on calibration using all available stations to ensure high reliability and accuracy.

### 3.4.3 Partial phase calibration for the integer multiples of $2\pi$

Besides testing the number of calibration points (Sect. 3.4.2), we also examine the impact of partial phase calibration. The calibration parameter ($C$) is considered to consist of two components: the integer multiple of $2\pi$ and the residual part less than $2\pi$. Three cases are tested to investigate the influence of different components of the calibration parameter on the $\Delta$SWE retrieval: no phase calibration, calibration using only the integer multiple of $2\pi$, and calibration using the full parameter. The second strategy means that only the integer multiple of $2\pi$ within the calibration parameter is used, i.e., full calibration

parameter subtracts its modulo $2\pi$. Specifically, if $C$ is with $(-\pi, \pi)$, $C = 0$; if $C \leq -\pi$, it is replaced by $-2\pi$; if $C \geq \pi$, it is replaced by $2\pi$. This approach removes the integer phase while ignoring the residual phase.

## 4 Results

This section shows the unwrapped phase and coherence of each InSAR pair in two snow seasons firstly. Then the retrieved time series scene-wide cumulative SWE is described, followed by the validation results, which compare retrieved and in-situ

$\Delta$SWE, and then assess the impact of using different numbers of calibrated points on validation accuracy. Finally, the validation of cumulative SWE and the comparison between the in-situ and retrieved cumulative SWE at each station are shown.

### 4.1 Intermediate results of InSAR processing

During the InSAR processing (Sect. 3.2), some intermediate results are obtained, including the unwrapped phase and coherence of each InSAR pair in two snow seasons, as shown in Figs. 5 and 6. During the snow season, some InSAR pairs show relatively

high coherence, corresponding to the subfigures 7 to 10 in Fig. 5, and 6 to 11 in Fig. 6. These relatively good interferometric pairs will be mentioned in the subsequent validation of results in Fig. 11. The areas with higher coherence correspond to places with more robust and accurate unwrapped phases. Lower coherence corresponds to more discontinuities in unwrapped phase

distributions, and more isolated areas appear in the connected components identified during the unwrapping process. These areas may increase the errors in the inversion results and result in adverse effects.

Based on the SWE and air temperature time series changes observed at the in-situ Wuxilike site (see Fig. 7), several patterns can be identified for the snow season. As shown in Fig. 7(a), a snowfall event was recorded in September 2019, but the corresponding interference pair dates (subplot 0 of Fig. 5 ) correspond to the period before and after the snowfall, showing little impact on the interference pair by snowfall and snowmelt. Then, up until mid-October, no snowfall is observed. During this time, changes in phase and coherence are likely caused by atmospheric variations driven by gradually decreasing

temperatures, as indicated in subplots 1 and 2 of Fig. 5, in which decorrelation and obvious changes in the unwrap phase begin to occur. Snowfall starts in mid-October, leading to a continuous increase in SWE. This results in a large area of low coherence, primarily due to the impact of snowfall, as shown in subplot 3 of Fig. 5. In early November, temperatures rise above 0 °C, leading to a snowmelt process. This causes coherence to remain low, as illustrated in subplot 4 of Fig. 5. A similar low coherence remains in subplots 5 and 6 of Fig. 5, this is likely due to heavy snowfall events and temperature fluctuations around

0 °C, causing an unstable snowpack state. The snowpack becomes more stable later in the snow season, accompanied by consistently low temperatures, mainly around −10 °C. During this period, coherence is relatively high, as shown in subplots 7 to 10 of Fig. 5. After mid-February of the following year, rising temperatures lead to a snowmelt process. The presence of wet snow significantly reduces coherence, as illustrated in subplots 13 to 17 of Fig. 5.

    We make a note here that, large snowfall events can cause temporal decorrelation, leading to phase unwrapping errors.

Under this condition, the SWE retrieval method is not recommended. For example, as shown in Fig. 7(a), the cumulative SWE increases by about 100 mm from 4 Nov 2019 to 28 Nov 2019 over one of the high-elevation stations, which exceeds the SWE changes detection limit (approximately 30 mm for C-band, corresponding to a $2\pi$ phase change). This suggests that phase ambiguity may occur. During this period, low coherence is shown by dark areas in subplots 5 and 6 of Fig. 5, which likely leads to unwrapping errors. These errors are visible as isolated unwrapped phase patches. Although our calibration constant is

designed to estimate and remove scene-wide phase bias (including integer multiples of $2\pi$), local unwrapping errors depend strongly on coherence and may persist even after calibration. To mitigate such effects, unwrapping correction, multilooking, and filtering are applied before calibration, which improves phase quality and minimizes the practical impact of phase ambiguities.

    The following year, a similar pattern is observed (Fig. 7(b)). The coherence is low from mid-to-late September to late

November due to the snowfall and the air temperature, which is not continuously below 0 °C. Lower coherence corresponds to larger unwrap phase changes. High coherence is recorded during low temperatures and stable SWE, as shown in subplots 6 to 11 of Fig. 6. In contrast, coherence decreases during the final snowmelt period when temperatures rise, as shown in subplots 12 to 16 of Fig. 6.

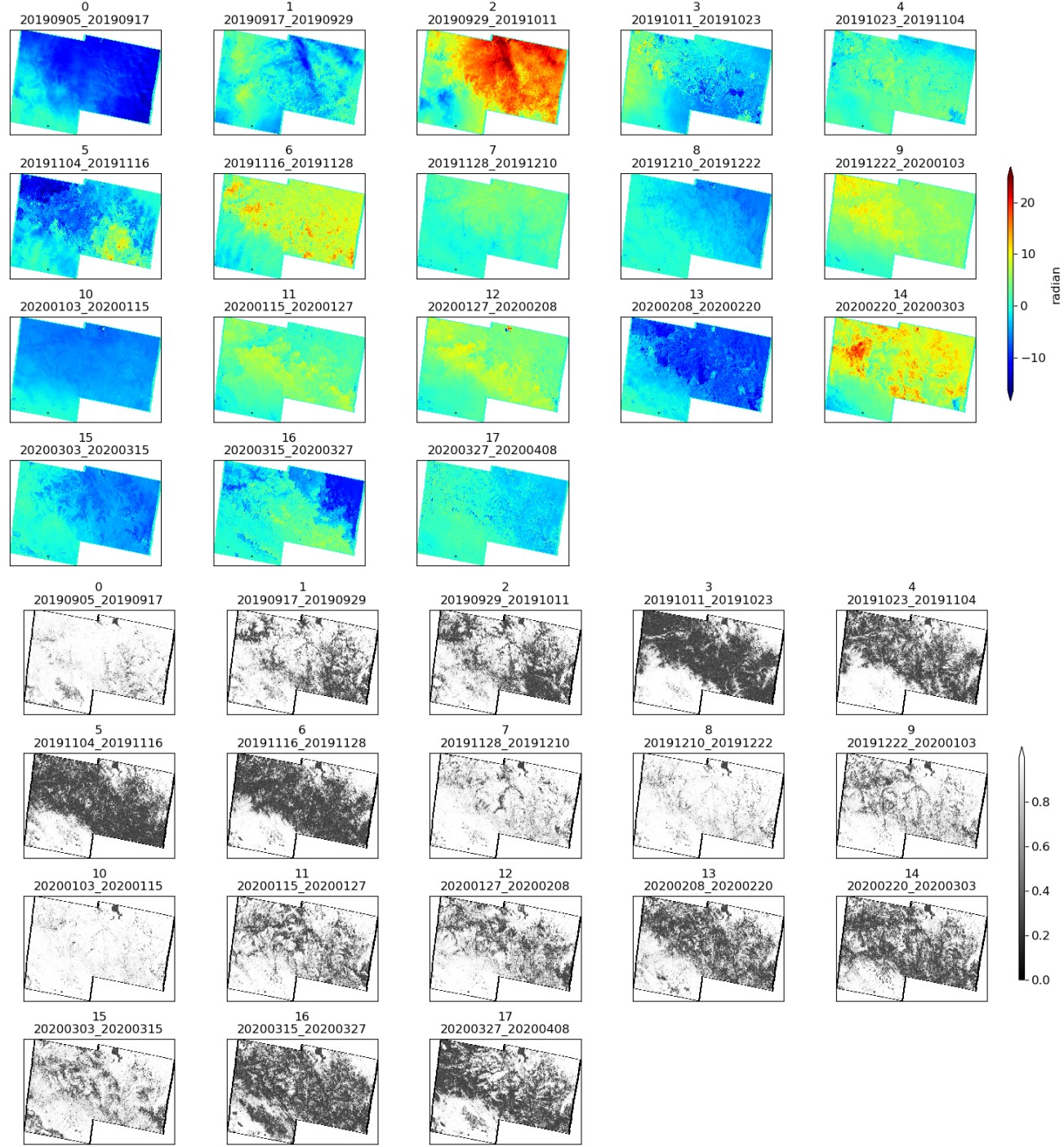

**Figure 5: The unwrapped phase (top panel) and coherence (bottom panel) data from September 5, 2019 to April 9, 2020, every 12 days.**


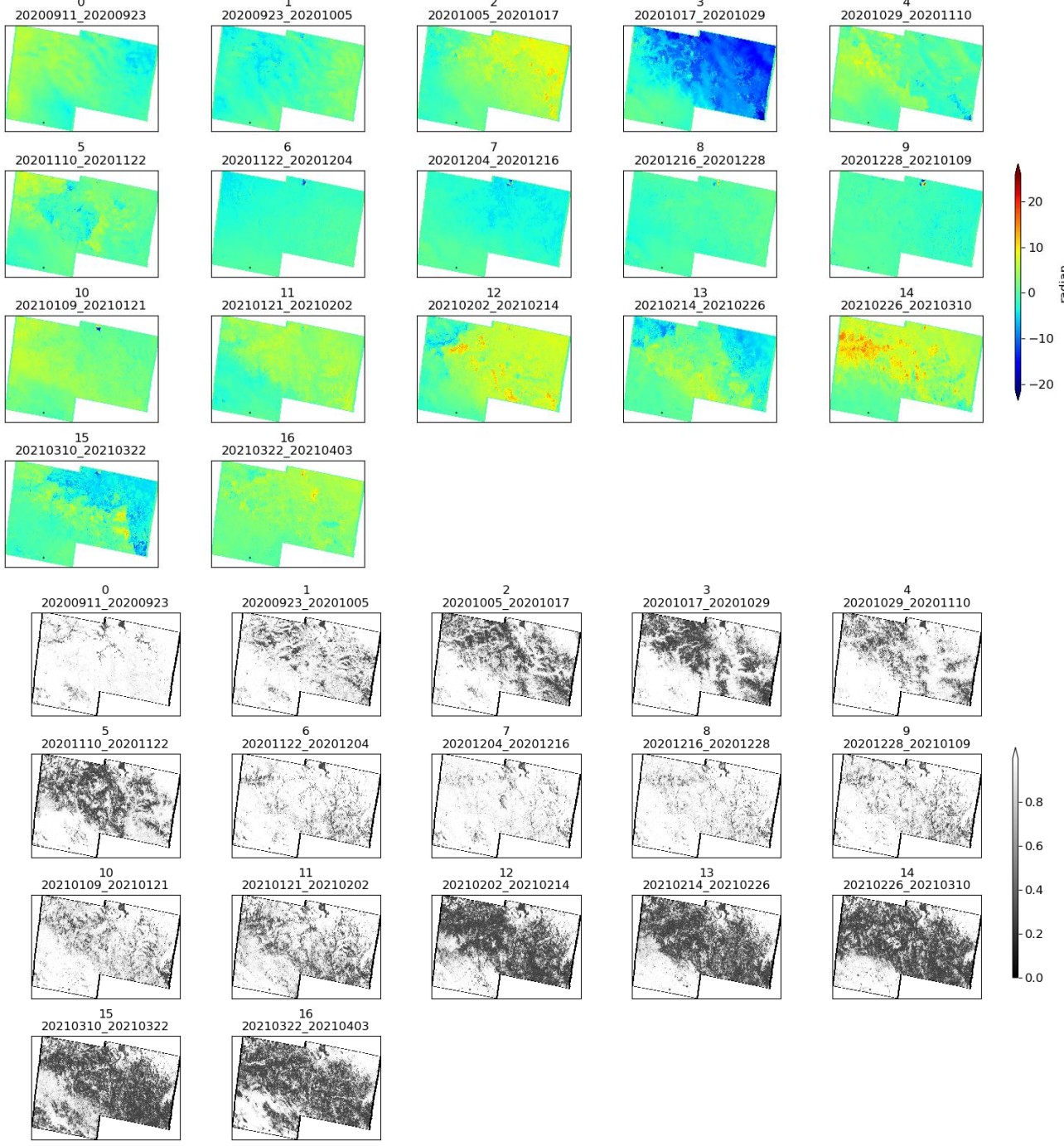

**Figure 6: The unwrapped phase (top panel) and coherence (bottom panel) data from September 11, 2020 to April 3, 2021, every 12 days.**

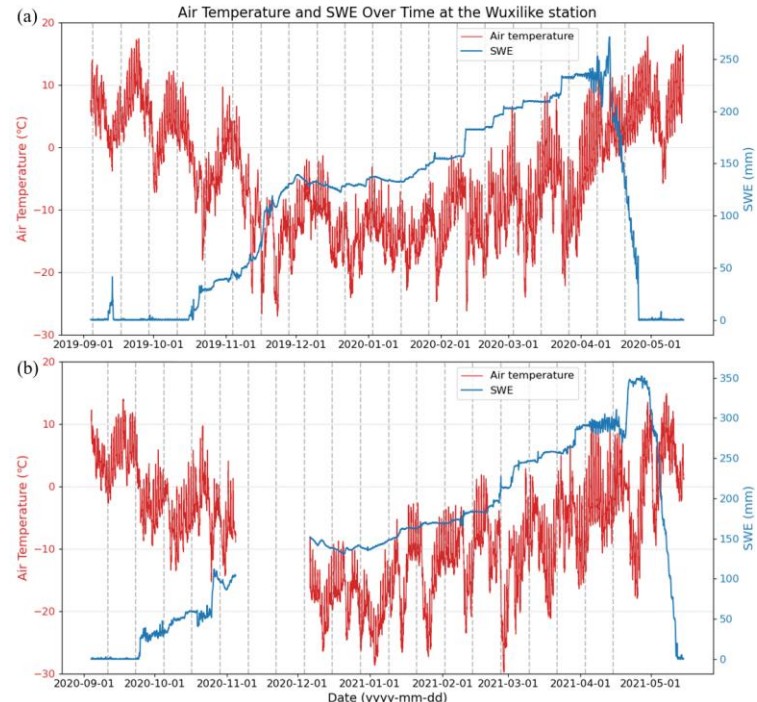

**Figure 7: Observed time series in-situ SWE and air temperature change at the Wuxilike station from (a) 2019 to 2020 and (b) 2020 to 2021. (The vertical black dashed lines correspond to satellite observation dates.)**


The decorrelation pattern in Figs. 5 and 6 (bottom panel) is related to topography. As shown in the red area of Fig. 1, the north-east region is characterized by relatively high elevations (above 2000 m), while the south-west part of the study area is lower (around 1000 m) and relatively flat. The observed north-west to south-east decorrelation coincides with the higher-elevation regions, where the complex topography likely contributes to reduced coherence. Moreover, the coherence pattern may reveal human activities (e.g., human grazing activities in September).


### 4.2 Spatiotemporal distribution in cumulative SWE

All of the time series retrieved ΔSWE are used to calculate the retrieved cumulative SWE at each date from the start date of our satellite's observation date, which can be expressed as

$$SWE(t_{i+1}) = SWE(t_0) + \sum_{t_j=t_0}^{t_i} \Delta SWE(t_j, t_{j+1}) \tag{14}$$

where $t_0$ is the start date (September 5, 2019 or September 11, 2020), $SWE(t_{i+1})$ is the cumulative (or absolute) SWE on the date of $t_{i+1}$, and $\Delta SWE(t_j, t_{j+1}) = SWE(t_{j+1}) - SWE(t_j)$. For example, the cumulative SWE at 20190929 (in yyyymmdd

format) is the summation of the cumulative SWE at the initial date of 20190905, ΔSWE(20190905,20190917), and ΔSWE (20190917,20190929).

Large spatial-scale cumulative SWE changes in the two snow seasons at the Altay, every 12 days, at a 75×100 m spatial resolution, are mapped using InSAR and Sentinel-1 following the above InSAR processing (see Figs. 8 and 9). A comparison of Figs. 8 and 9 show that the processes of spatial variation for accumulated SWE in both the 2019-2020 and 2020-2021 seasons are similar. In both seasons, the SWE shows a gradual increase from September to the end of March. However, differences are also observed. In the second year, the spatial extent of maximum SWE cumulation was smaller than in the first year. Additionally, the location and the range of the region where SWE reached its peak are different between the two years.

During the 2019–2020 snow season, SWE cumulation is in a moderate pattern in the early months, from September to November, SWE increases from 0 mm to approximately 50 mm in general. As the season gets into late January, significant cumulation is observed in areas A and B, located at higher elevations, while the lower-elevation area C in the southwest shows a relatively smaller increase. By the end of March, SWE reaches its maximum spatial extent, with the most notable cumulations still emerging in the higher elevations, areas A and B. This is also evident when comparing Fig. 8(16) and Fig. 8(17), where the SWE values in Fig. 17 become larger and the area covered by higher SWE values (red) expands. This increase is consistent with the in-situ SWE measurements at the Wuxilike station during the same period, as shown in Fig. 7(a).

A similar temporal process of SWE cumulation can be found in the 2020-2021 season, though with differences in the spatial variations. Early cumulation trends are similar to those of the previous year, with SWE rising to approximately 50 mm by late November. However, uneven increases appear across the study area from December to February, which may be explained by the influence of meteorological conditions and topographic factors. The rise in SWE is concentrated in area A, where the peak cumulation in spatial extent is reached on April 3, 2021, as shown in Fig. 9(17), with a smaller spatial extent of increase compared to the previous snow season. A rapid SWE decrease in late March is observed in area B, where SWE declines from approximately 100 mm to around 50 mm. These spatiotemporal differences reflect the uneven distribution or temporal dynamics of snow accumulation and melting processes. They are influenced by various factors during the snow season, such as differing snowfall patterns, topography(Eppler et al., 2021), vegetation(Georg et al., 2007), and meteorological conditions (Deeb et al., 2011).

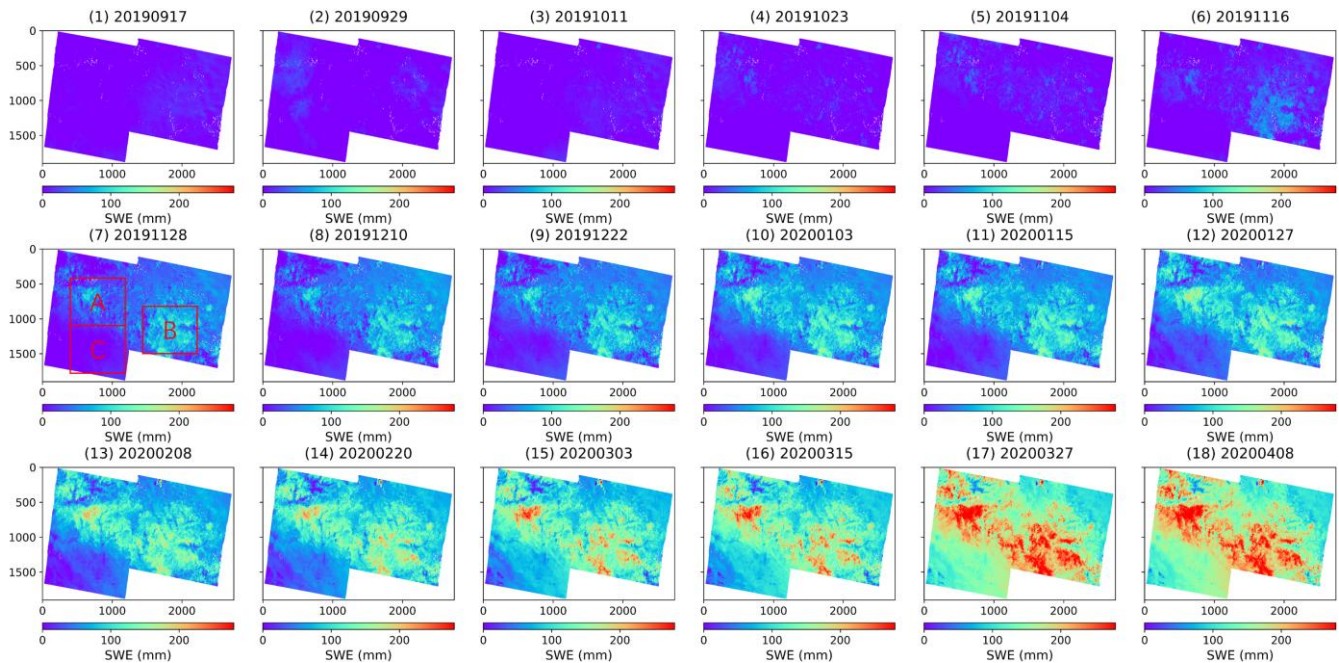

**Figure 8: Spatiotemporal distribution of cumulative SWE in Altay during the 2019–2020 snow season. (Both this figure and the following one show the SWE variation relative to the first reference scene, with a 12-day cumulation interval. The SWE of the reference scene is set to 0. The reference scene for this figure is September 5, 2019. The cumulation's end dates are shown at the top of each sub-figure. The red rectangles mark areas A, B, and C to describe the SWE variations across different regions. To improve comparison, these are colored in the same range.)**

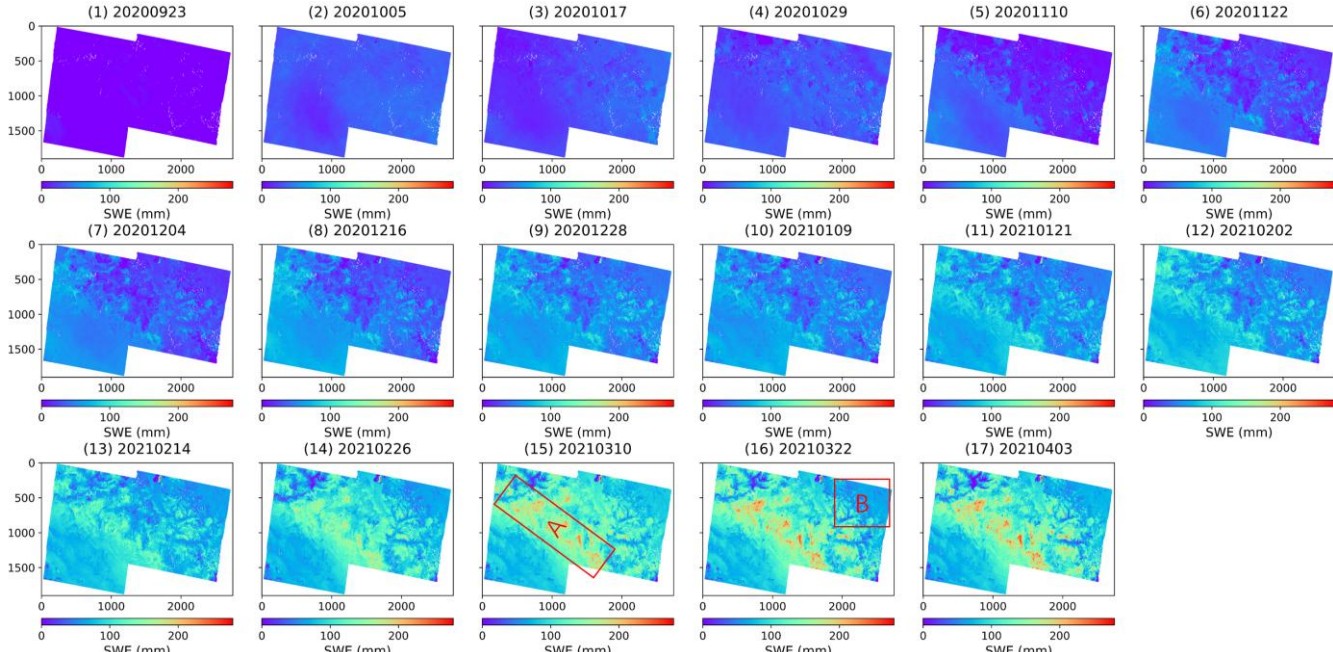

**Figure 9: Spatiotemporal distribution of cumulative SWE in Altay during the 2020–2021 snow season. (The reference scene for this figure is September 5, 2020.)**

### 4.3 Validation of the retrieved SWE

#### 4.3.1 Comparing Sentinel-1 retrieved *ΔSWE* with in-situ *ΔSWE*

As shown in Fig. 10, the retrieved $\Delta SWE$, with a 12-day temporal baseline from September 5, 2019, to April 8, 2020, and
September 11, 2020, to April 3, 2021, are validated against all in situ $\Delta SWE$ observations, with an RMSE of 9.5 mm (R =
0.56, p-value << 0.05). Here, we use a 5×5 pixel averaging on retrieved $\Delta SWE$, corresponding to a spatial resolution of
375×375 m. The high coherence points (red) are closer to the 1:1 line with higher accuracy, while the lower coherence ones
(purple) are affected by the decorrelation error sources and thus more scattered away from 1:1 line. These results prove that
InSAR-derived $\Delta SWE$ using Sentinel-1 with a 12-day revisit time is able to estimate the $\Delta SWE$ at Altay, indicating that the
importance of high coherence is one of the key factors to obtain good retrieval results. The validation results show minor
differences depending on the choice of multi-pixel averaging for the retrieved ΔSWE. Notably, our results are close to those
in Idaho (RMSE = 7.6 mm, R = 0.82; Oveisgharan et al., 2024), although their study used a shorter 6-day temporal baseline,
whereas ours relied on 12 days.

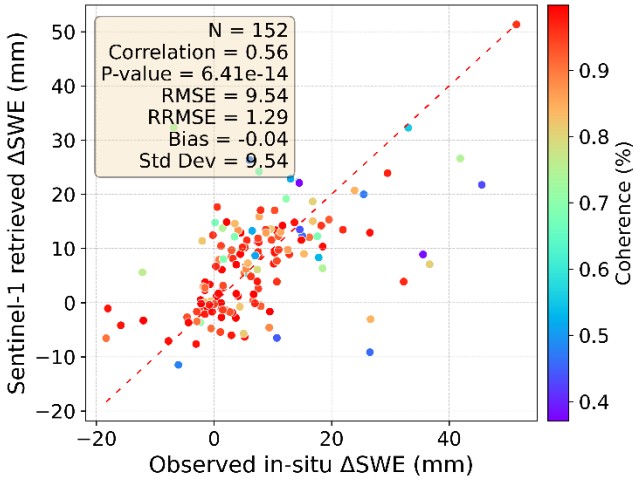

**Figure 10: Comparison between the in-situ *ΔSWE* in 12 days and the Sentinel-1 InSAR retrieved *ΔSWE* changes at Altay in 2019-2021. (The color of the points corresponds to the coherence.)**

As illustrated in the previous section, there are some differences in spatiotemporal cumulative SWE in the two snow seasons.
To examine whether the differences in spatiotemporal cumulative SWE between the two seasons affect retrieval accuracy, the
total validation results are divided by year, with each time series InSAR pair assessed separately (see Fig. 11). The validation
results remain reliable, with similar performance when analyzed separately for each snow season or combined, indicating the
stability of the retrieval method across different snow seasons.

The retrieved $\Delta SWE$ are validated against in-situ $\Delta SWE$ with an RMSE of 10.1 mm (R = 0.58, p-value << 0.5) in 2019-
2020, 8.6 mm (R = 0.48, p-value << 0.5) in 2020-2021. The points that belong to higher coherence (circled) InSAR pair are
closer to the 1:1 line, showing a good agreement in the retrieved and in-situ $\Delta SWE$.

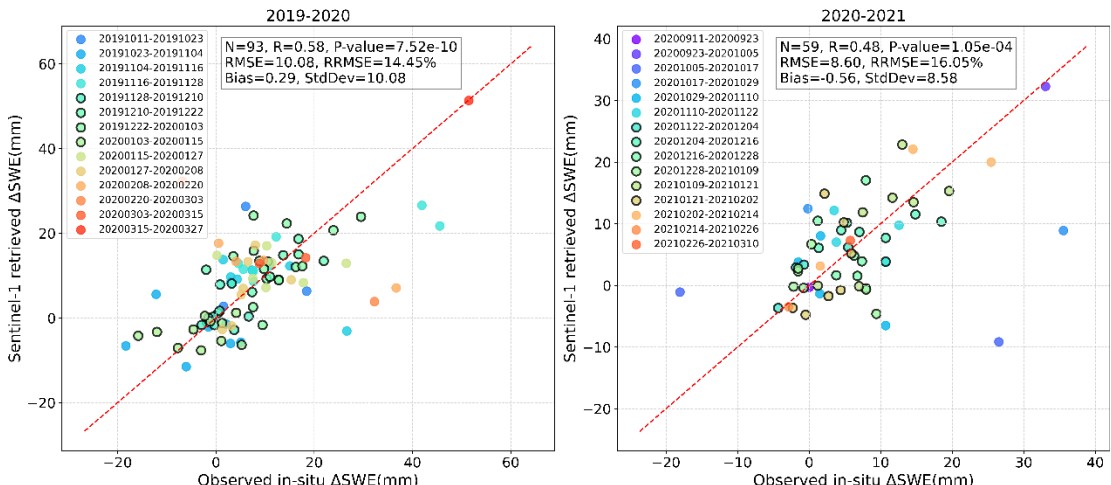

**Figure 11: Comparison between the in-situ ΔSWE changes in 12 days and the Sentinel-1 InSAR retrieved ΔSWE changes at Altay**
**in the snow seasons of (a) 2019-2020 and (b) 2020-2021. (The color of the points corresponds to each InSAR pair. The points with**
**high coherence in the whole InSAR pair scene are circled.)**

### 4.3.2 Validation results under different numbers of selected points

Based on the method introduced in Sect. 3.4.2, part of the $\Delta SWE$ observations in each InSAR pair in 2019-2020 are used to
derive the calibration parameter. Then, the rest of $\Delta SWE$ observations are used to validate the retrieved ΔSWE after calibrating
each InSAR pair using the derived calibration parameter. As shown in Fig. 12, the maximum number of available ΔSWE
observations in all InSAR pairs in the 2019-2020 snow season is 12. Hence, the maximum number of points used for calibration
is limited to 9 to ensure that at least 3 points remain for validation. One hundred tests based on the random selection of stations,
varying the number of points used for calibration, are carried out, and the following validation results are shown in Figs. 13
and 14.

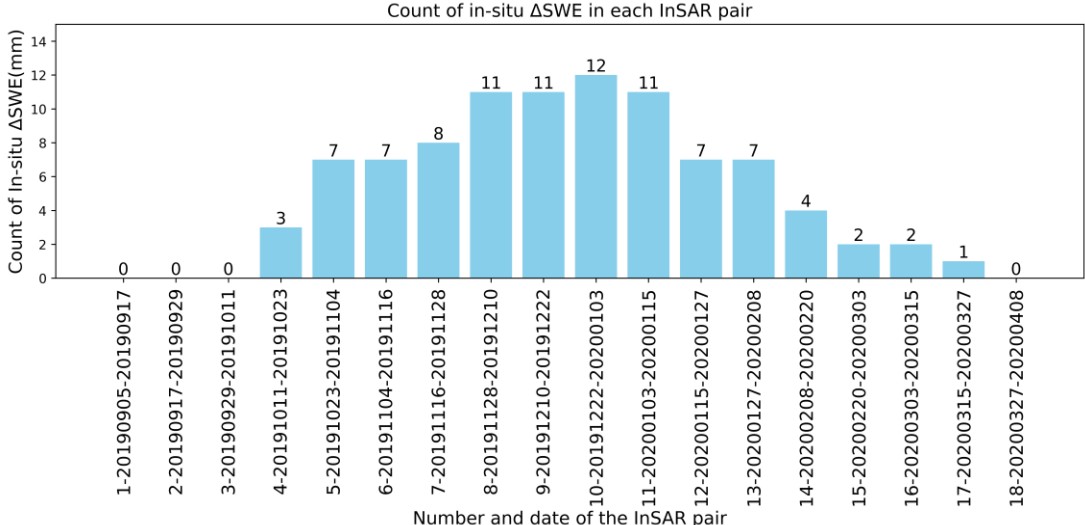

**Figure 12: The number of available in-situ ΔSWE in each InSAR pair in 2019-2020**

Validation results based on calibration parameters computed using coherence-weighted least squares, under different numbers of randomly selected calibration stations after 100 Monte Carlo realizations, are presented in Fig. 13. The validation shows that accuracy changes as the number of points changes. Even using a single calibration station can produce reasonably good retrievals, with an average R of 0.46 and RMSE of 12 mm. In particular, accuracy improves as the number of calibration stations increases from 1 to 6 at first, then stabilizes when the number of calibration points varies between 6 and 8. However,
as the number of calibration points increases beyond 8, the rapid reduction in validation points is likely the main reason for a degraded validation performance.

  This shows that selecting approximately half of the available stations (i.e., 6 out of 12) for calibration yields reliable validation results. In general, increasing the number of calibration points improves the accuracy; however, when too many points are used, the number of remaining validation points decreases rapidly, which can adversely affect the validation statistics.
Therefore, all other retrieval results presented in this study are based on calibration using all available stations to ensure high reliability and accuracy.

  This trend is further illustrated in a scatter plot comparing the in-situ ΔSWE with the Sentinel-1 InSAR-derived ΔSWE for one representative realization out of the 100 trials under different numbers of selected calibration stations (from 1 to 9), see Fig. 14. The results show good validation performance when using 6, 7, or 8 calibration points. This finding also suggests that
selecting at least half of the available ΔSWE values for calibration can yield reliable InSAR-derived ΔSWE estimates.

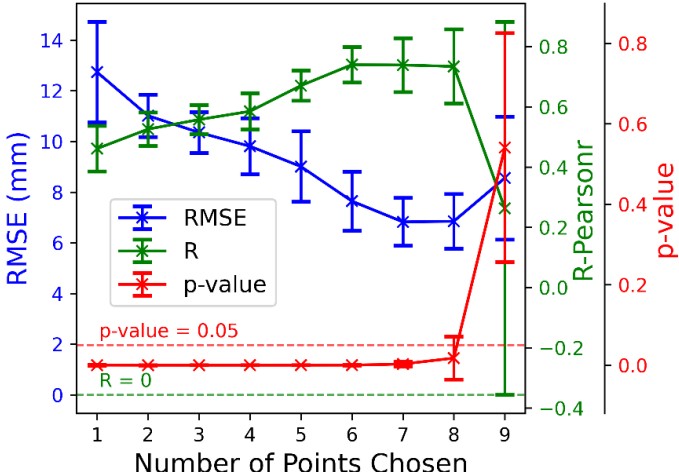

**Figure 13: Variation of RMSE, Pearson correlation, and p-value with the number of points used for calibration after 100 Monte Carlo random point selection tests. (The two ends of the long line represent the range, and the '×' symbol in the middle represents the average value over 100 times.)**

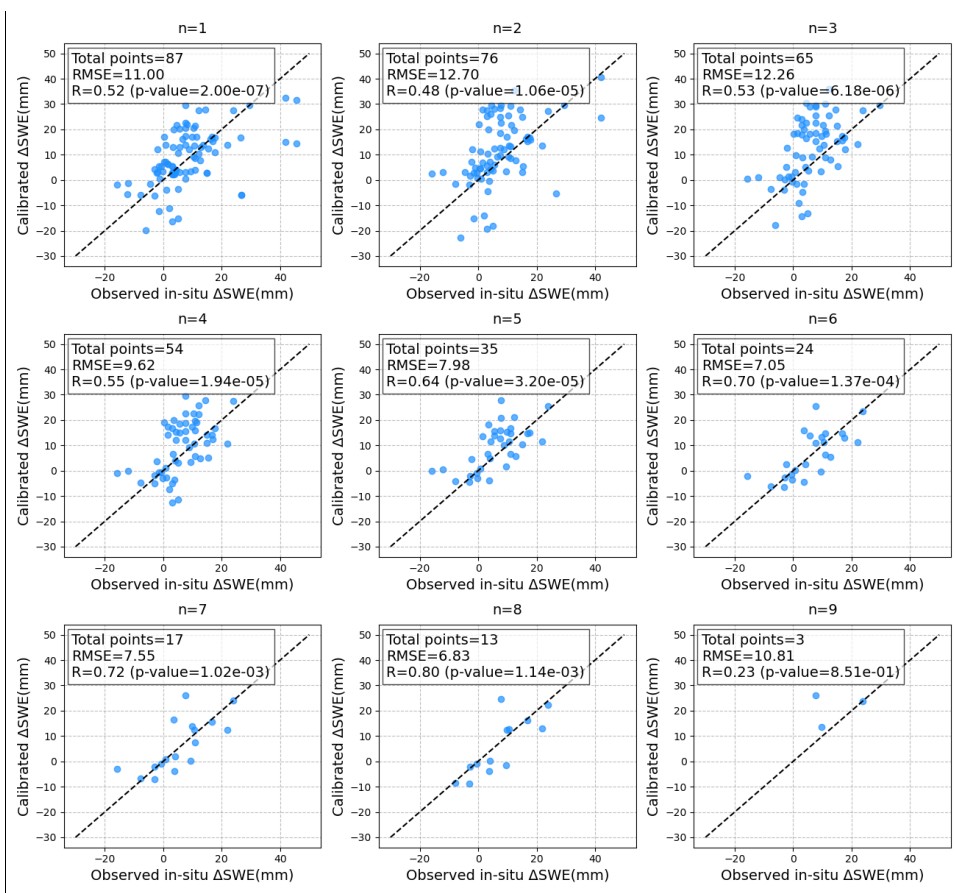

**Figure 14: Comparison between the in-situ $\Delta SWE$ and the Sentinel-1 InSAR retrieved $\Delta SWE$ under different numbers of selected calibration points, from 1 to 9, in one realization.**

## 4.4 Comparing Sentinel-1 retrieved cumulative SWE with in-situ cumulative SWE

It should be noted that discrepancies in initial SWE values may exist when comparing cumulative SWE from in-situ measurements and that derived from InSAR observations. To ensure a consistent basis for comparison, an initial value alignment is performed before the validation analysis. On the one hand, the satellite-derived SWE is referenced to the first Sentinel-1 acquisition, where the initial SWE is set to zero. However, at the same time, a few in-situ stations may record small but nonzero SWE values due to early snowfall events. On the other hand, for most stations, no snowfall is recorded on the date of the first acquisition or even several subsequent acquisitions, so the measured cumulative SWE remains zero. In contrast, the InSAR-derived cumulative SWE may show nonzero values at the same acquisition due to atmospheric effects or other factors accumulating over time. As a result, inconsistencies in initial SWE values may also occur at later dates, even when no snowfall is observed. This discrepancy introduces a constant offset when comparing the retrieved cumulative SWE with in-situ data. To eliminate this offset and ensure consistency in the initial comparison, the satellite-derived cumulative SWE is adjusted to match the first available in-situ cumulative SWE observation on the same date. This alignment procedure is then applied to all subsequent satellite-derived cumulative SWE values.

### 4.4.1 Validation of retrieved cumulative SWE

As shown in Fig. 15, the retrieved cumulative SWE is validated against in situ SWE (referring to cumulative observations) after excluding wet snow, with an RMSE of 40.9 mm (R = 0.65, p-value $\ll$ 0.05). We clarify that for $\Delta SWE$, wet-snow periods were explicitly excluded and only dry-snow conditions were retained, following the criteria described at the end of Sect. 3.1. However, since cumulative SWE is obtained by integrating the $\Delta SWE$ time series, continuity of the full snow season is required for comparison. Therefore, some wet-snow points were temporarily retained in the cumulation process (shown as light grey dots in Fig. 15 and the cumulative SWE curves in Figs. 16−17), but these were excluded in the final cumulative SWE evaluation.

After this exclusion, some underestimated and scattered points deviate significantly from the 1:1 line. Most of these points correspond to high-elevation stations such as Wuxilike, Wuxilike-Muban, and Tollheit (not shown due to limited valid data). We excluded these stations from further validation. The locations and elevations of these stations can be found in Fig. 2, and the cumulative SWE underestimation is illustrated later in Fig. 17 and Fig. 18. Tollheit is not shown due to the limited number of valid data points. Specifically, heavy snowfall (around 100 mm) at the Wuxilike station (and nearby Wuxilike-Muban) in early Nov 2019 lead to low coherence and phase unwrapping errors (as shown in Fig. 5 and Fig. 7). This leads to $\Delta SWE$ underestimation errors propagating through the time series. Improved validation results are obtained by excluding these underestimated points, with an RMSE of 28.3 mm and R=0.70.

The bimodal scatter distribution observed in the validation results is mainly attributed to the error propagation in the time series retrieved $\Delta SWE$. Since cumulative SWE is calculated by summing $\Delta SWE$ from consecutive interferometric pair, any overestimation or underestimation of $\Delta SWE$ in a single pair propagates through the subsequent cumulative SWE, leading to

deviations (to be illustrated in detail in Fig. 22 and Fig. 23). As a result, the scatter tends to split around the 1:1 line, forming a bimodal pattern. Nevertheless, despite this apparent bimodal scatter distribution. In general, the higher in situ SWE values generally correspond to higher retrieved SWE values, which means the overall trend of the retrieved cumulative SWE remains

consistent with the in-situ measurements.

Here, the wet snow points (light grey) are excluded, as they meet the wet-snow exclusion criteria described at the end of Sect. 3.1. This exclusion is primarily due to the limited penetration depth of C-band radar signals in wet snow, which restricts the ability to retrieve reliable SWE estimations under such conditions. To better understand this limitation, the physical basis of wet snow interaction with radar signals is briefly discussed below.


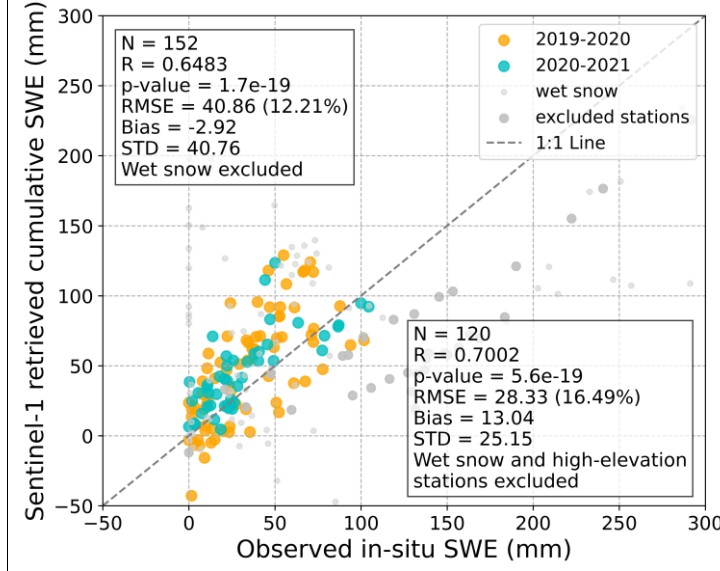

**Figure 15: Comparison between the Observed in-situ SWE and the Sentinel-1 InSAR retrieved cumulative SWE at Altay in 2019-2021.**

The penetration depth of a medium $\delta_p$ is related to the volume absorption coefficient $\kappa_a$ as

$$\delta_p = \frac{1}{\kappa_a} \tag{15}$$

where $\kappa_a$ is related to the effective dielectric constant of the wet snow $\varepsilon_{ws}$ as

$$\kappa_a = \frac{4\pi}{\lambda} Im\{\sqrt{\varepsilon_{ws}}\} \tag{16}$$

The permittivity of wet snow at C-band (5.405GHz) can be given by the following equations from the modified Debye-like model (Hallikainen et al., 1986):

$$\varepsilon_{ws} = \varepsilon'_{ws} - j\varepsilon''_{ws} \tag{17}$$

$$\varepsilon'_{ws} = 1 + 1.83\rho_s + 0.02m_v^{1.015} + 0.0539m_v^{1.31} \qquad (18)$$

$$\varepsilon''_{ws} = 0.0321m_v^{1.31} \qquad (19)$$

where $\rho_s$ is the snow density (g/cm$^3$), and $m_v$ is the volume fraction of liquid water in the snow mixture (%).

As shown in Fig. 16(A), significant differences exist in the interaction mechanisms between electromagnetic waves and dry snow versus wet snow. Electromagnetic waves interact primarily with the surface layer of wet snow, resulting in an increase of the scattering phase center compared to dry snow. This leads to a loss of coherence between the wet snow signal and the previous snow-free observation. Particularly at the end of the snow season, the C-band electromagnetic waves can not penetrate the snow to the ground with the increase of $m_v$ caused by the snow melt process. For example, in Fig. 16(B), a penetration

depth $\delta_p$ of approximately 5 cm is observed when $m_v$ is 6%, but usually the snow depth is larger than 20 cm. Under these conditions, errors will be introduced if the retrieval algorithm Eq. (3) for dry snow scenarios is applied. Therefore, the wet snow data are excluded during the validation. Nevertheless, when the liquid water content is very low in the early stages of snowmelt (e.g., $\delta_p = 33$ cm for $m_v = 1\%$), the radar signal still can penetrate most of the snowpack, allowing the impact of melting on SWE reduction to be effectively captured. It should also be noted that the failure of the inversion algorithm occurs

gradually.

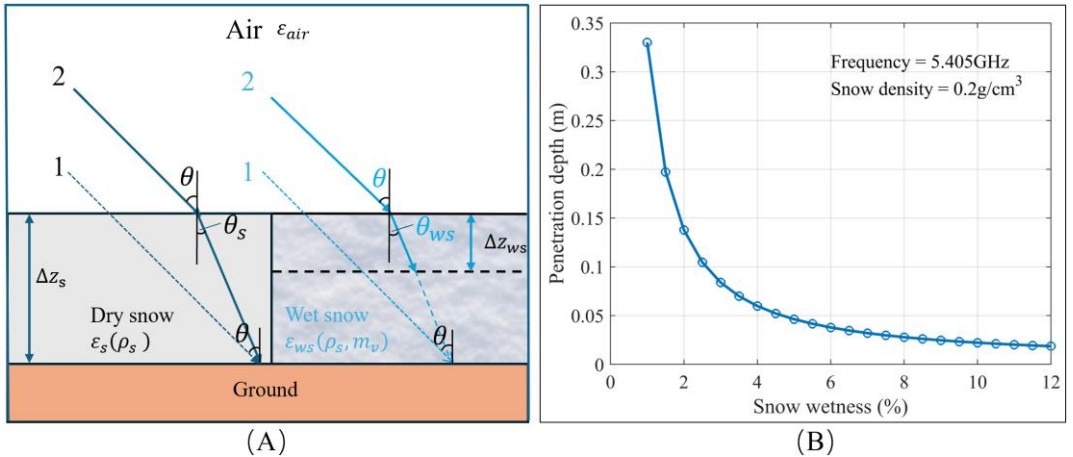

**Figure 16: (A) Propagation path of radar wave in dry snow and wet snow, and (B) penetration depth of wet snow at the frequency of 5.405 GHz and the snow density of 0.2 g/cm³.**


**4.4.2 Comparing Sentinel-1 retrieved time series cumulative SWE with in-situ cumulative SWE**

The time series of retrieved cumulative SWE is evaluated at each in-situ SWE station (Figs. 17 and 18, for each year, respectively). Good agreements (except for the period after mid-March) are shown between the in-situ and retrieved cumulative SWE at some stations, such as stations (2), (6), and (10) in Fig. 17 and station (4) in Fig. 18. Nevertheless, an overestimation

of about 20 to 60 mm is exhibited by the satellite retrieval at the Dadonggou and Altay station (i.e., the blue in-situ SWE line plots below the others) in two snow seasons in Fig. 17(9) and Fig. 18(2), Fig. 17(12) and Fig. 18(5), respectively. In contrast, the Wuxilike station is underestimated by about 100 mm in December (the blue in-situ SWE line plots above the others) in two snow seasons in Fig .17(8) and Fig. 17(1)). These overestimations and underestimations for two consecutive years may be related to the inherent environment of each site. For example, underestimation occurs at the Wuxilike station, located in

relatively high elevations (around 2146 m), while overestimation is observed at stations (13) and (17), situated in lower elevations (around 1076 m and 730 m, respectively). These differences may be attributed to variations in local slope, surface conditions, and nearby mountainous areas, which together affect the interferometric phase signal and introduce biases in the retrieved SWE. At the Wuxilike station (Fig. 17(8)), heavy snowfall is recorded from early November to early December. Although an increasing trend is captured in the retrieval, a notable underestimation remains, owing to decorrelation and phase

unwrapping errors caused by heavy snowfall. This time-series result is consistent with the above temporal decorrelation analysis (Fig. 5 and Fig. 7) as well as the underestimation of the cumulative SWE in Fig. 15 (the large light gray dots).

      However, some sites show different estimations (overestimation in one year and underestimation in another year) in two years. This phenomenon may be related to the different snow accumulations over the two years. Another reason might be that the errors will accumulate as time passes. This means one pair of overestimations and underestimations will propagate on the

following estimations. Moreover, the overestimated station may become underestimated after the phase calibration process, or vice versa. The choice of pixel averaging can affect the final retrieval results, too. For example, at the Xiaodonggou station in Fig. 17 (10), the retrieval accuracy improves as the averaging window size increases. This may be related to the mountainous terrain rather than the uniform plain surrounding the station, where a larger averaging window may have more impact on the retrieval results. In addition, for Fig. 17, we observe that at several stations (6, 7, 9, and 10), the in-situ SWE measurements

decrease during mid-to-late March, while the satellite-derived SWE continues to increase. However, most of these retrievals follow a pattern consistent with the in-situ SWE observed at the Wuxilike-Muban station (blue curve in Fig. 17(11)). This is probably because the Wuxilike-Muban station was used in the estimation of calibration parameters with the coherence-weighted least squares method, and its observations generally show high coherence.

      Despite some overestimation and underestimation, the retrieved SWE trends are generally consistent with in-situ

measurements across all stations, which agrees with the findings of Oveisgharanet al. (2024), who suggest that the main reason for these discrepancies is likely related to phase unwrapping errors and phase ambiguities. In addition, we found that using the coherence-weighted least squares method generally provides better results than the ordinary least squares method. Moreover, we found that using the coherence-weighted least squares method generally provides better results than the least squares method (not weighted), as indicated by the smaller differences between the in-situ and retrieved cumulative SWE in Figs. 17

and 18.

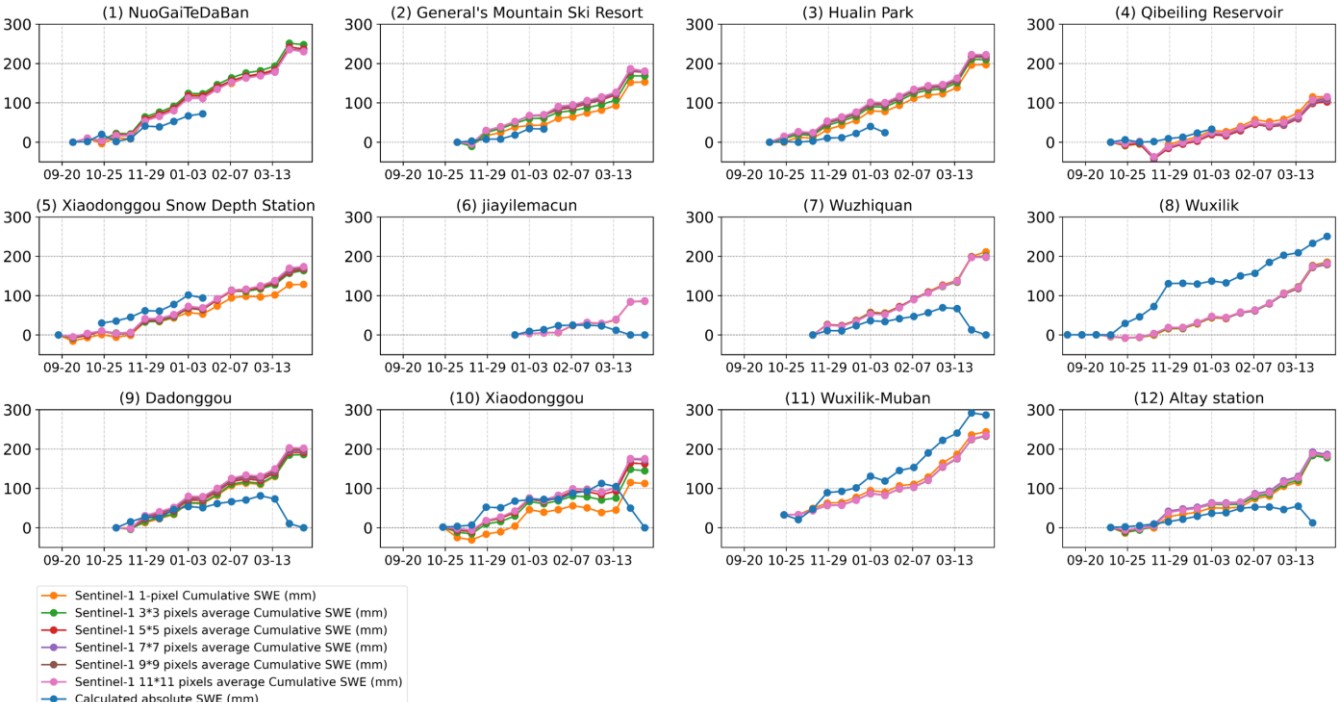

**Figure 17: Time series of in-situ and retrieved cumulative SWE using Sentinel-1 interferometric phase for different stations in Altay during the 2019-2020 snow season. (The in-situ cumulative SWE is represented by blue lines, while the retrieved cumulative SWE is shown by six different colored lines, each corresponding to a different spatial scale. Specifically, the retrieved SWE values are calculated as the average SWE within a window centered at the station's latitude and longitude, with window sizes of 1×1, 3×3, 5×5, 7×7, 9×9, and 11×11 pixels, respectively. The number in the upper right corner is the elevation of the site. The same color scheme apply to the following Fig. 18 for the 2020–2021 snow season.)**

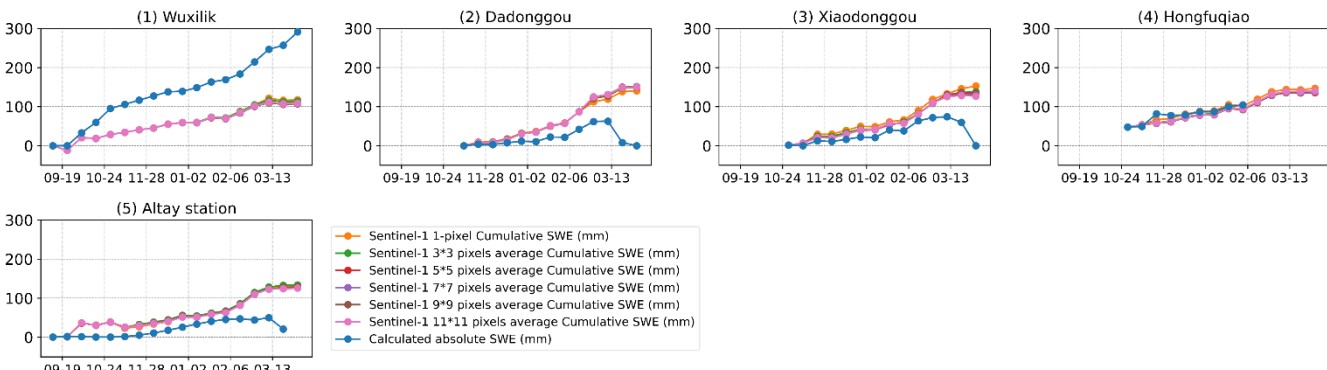

**Figure 18: Time series of in-situ and retrieved cumulative SWE using Sentinel-1 interferometric phase for different stations in Altay during the 2020-2021 snow season.**

**5 Discussion**

Good inversion performance is demonstrated in Sect. 4 in general, but some inconsistencies between retrieved and measured SWE can be seen, which are yet to be further investigated. For example, the points which are far from the 1:1 line in Fig. 10, and the overestimation and underestimation of retrieved cumulative SWE in Figs. 15 and 16. In this section, we explore why different points show varied accuracy and what the optimal occasions of this retrieved method are by using available data on meteorology and snow properties. Then, the validation of cumulative SWE and $\Delta SWE$ at the same site is shown next with the scatterplot. At the end of this section, the effects of partial phase calibration on the inversion result are studied.

**5.1 Analysis of multiple factors influencing the retrieval of $\Delta SWE$**

This section investigates the influence of various factors, including coherence, air temperature, elevation, slope, snow depth, and snow density, on the validation of retrieved $\Delta SWE$, as illustrated in Fig. 19. Those with lower coherence (purple) become more scattered, as shown in Fig. 19(1). As shown in Fig. 19(2), the lowest temperature (purple) values cluster around the 1:1 line. Points at lower elevations (Fig. 19(3)) appear more accurate, and red points at high altitudes tend to show an underestimation. For Fig. 19(5) and (6), the purple points corresponding to lower values of snow depth (about 15 cm) and snow density (about 150 kg/m³) appear more concentrated compared to higher values.

Furthermore, to better understand the relationship between coherence, topography, and in-situ $\Delta SWE$, additional relationships are illustrated in Fig. 20. A weak negative correlation is observed between coherence and in-situ $\Delta SWE$ (R = −0.32) in Fig. 20(A). Most points cluster at high coherence (> 0.8) within a narrow $\Delta SWE$ range (–10 to 20 mm), while a few with $\Delta SWE$ > 30 mm drop to lower coherence (< 0.6), leading to the weak overall correlation. The reduced coherence observed at $\Delta SWE$ > 30 mm may be related to phase unwrapping, since the phase-to-$\Delta SWE$ conversion (Eq. (3)) indicates that one $2\pi$ cycle corresponds to about 30 mm of $\Delta SWE$, beyond which phase unwrapping errors can occur.

In addition, elevation shows a weak positive correlation with in-situ $\Delta$SWE (R = 0.29) in Fig. 20(B). This reflects that higher elevations generally experience heavier snowfall and accumulate thicker snowpacks, leading to larger SWE changes. Moreover, coherence and elevation are negatively correlated (R = −0.34). At higher elevations, coherence values exhibit a wider distribution, including both high and low values, whereas lower elevations tend to cluster at higher coherence levels. This pattern, also visible in Fig. 19(1) and (3), may be associated with larger snowfall at higher elevations, which could contribute to the reduced coherence. This is consistent with the findings that large increases of SWE can create ambiguities in its retrieval by inducing phase wrapping (Engen et al., 2003; Ruiz et al., 2022).

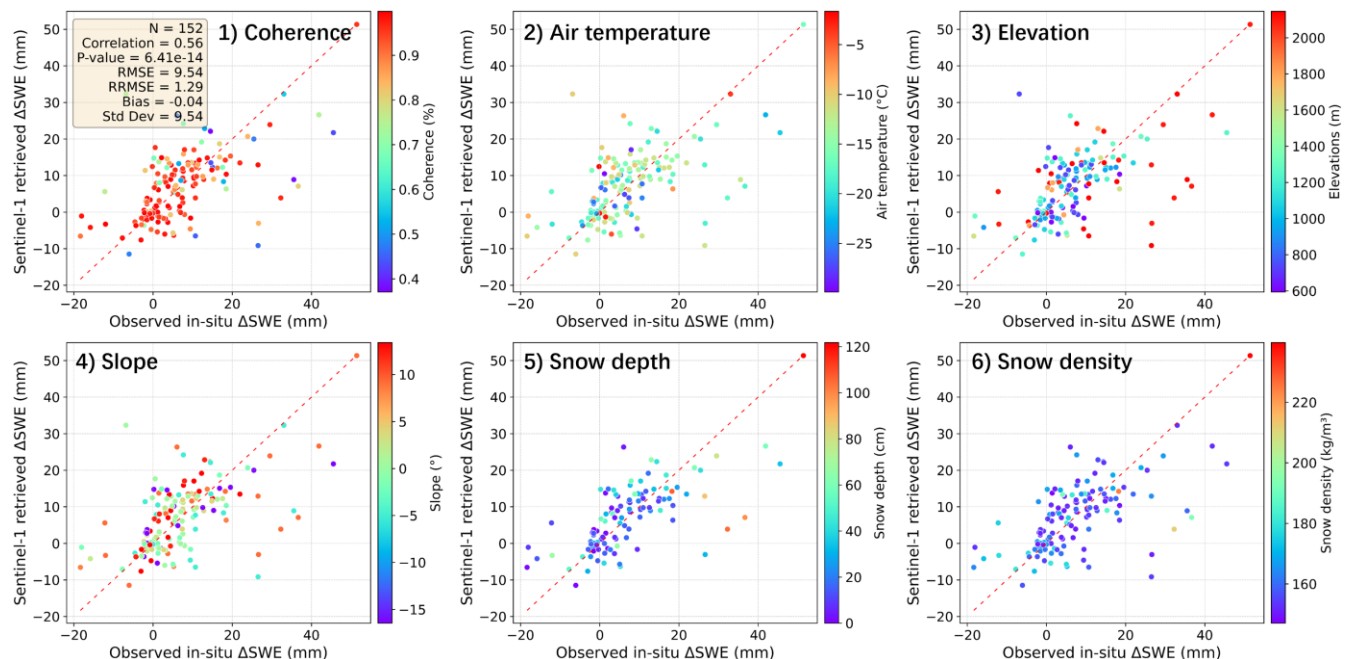

**Figure 19: Analysis of multiple factors influencing the retrieval of ΔSWE. (Each sub-figure is colored based on different properties)**

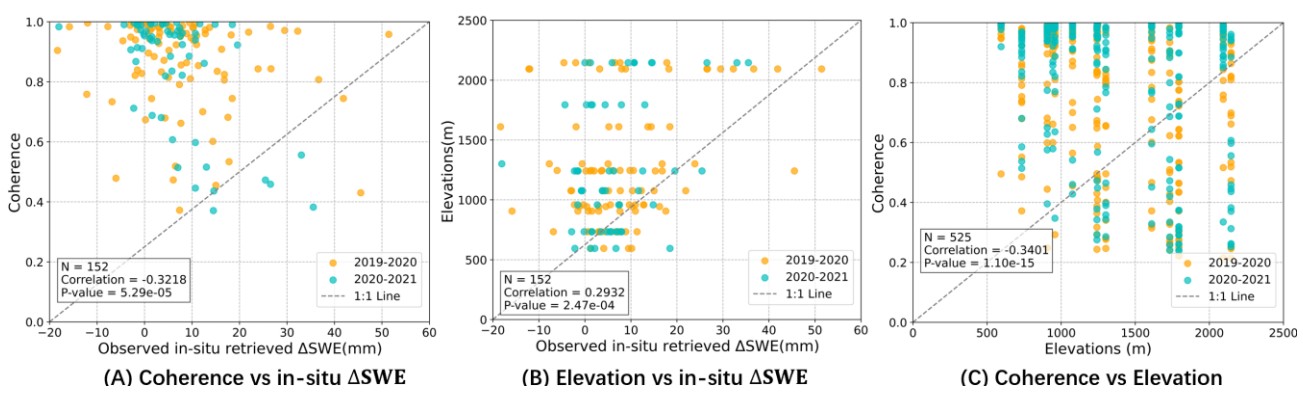

**(A) Coherence vs in-situ ΔSWE**   **(B) Elevation vs in-situ ΔSWE**   **(C) Coherence vs Elevation**

**Figure 20: Analysis of coherence and topographic effects on in-situ Δ𝑆𝑊𝐸. (A) Coherence vs. in-situ ΔSWE, (B) Elevation vs. in-situ Δ𝑆𝑊𝐸 and (C) Coherence vs. Elevation.**

Additionally, data is removed based on specific thresholds for each factor to assess the impact on inversion results, as summarized in Table 1. For coherence, the threshold for removing low values was gradually increased from 0.4 to 0.7 (with no points having coherence below 0.2). After removing low-coherence points, inversion results show minimal changes, with a slight decrease in RMSE and a slight increase in correlation. This may be because only 16% of points have coherence below 0.7, and more than half (62%) have coherence above 0.9, see Fig. 21(a). When filtering higher temperature values (from 0 °C to −20 °C, decreasing by 5 °C), significant improvements are observed after removing points above −20 °C. The correlation increases to 0.75, and RMSE decreases to 8.14 mm. These points, corresponding to temperatures below −20 °C, are from four InSAR pairs collected from November to January during the dry snow season. For elevation and slope, the inversion results

after limiting these ranges may not be entirely reliable, as the data distribution is not balanced (see histograms in Fig. 21 (c) and (d)), and the total number of data points is relatively small. Elevation and slope are intrinsic to the site, meaning each station corresponds to specific values of elevation and slope. Regarding snow depth, we test removing points with thick and shallow snow depths, retaining points with snow depths in a certain range, and removing thick snow (from 20 cm to 140 cm, increasing by 20 cm). The lowest RMSE is obtained by retaining only points with snow depths between 0 cm and 20 cm, where the R-value reaches 0.57 and the RMSE is about 7 mm. This relatively good performance may be related to the fact that errors tend to be smaller under shallow snow conditions. This shallow snow (between 0 cm and 20 cm) constitutes a large proportion of the dataset (37%, as shown in Fig. 21) compared to other depth ranges. For snow density, limiting the snow density in the scatter plot to between 150 kg/m$^3$ and 200 kg/m$^3$ will slightly improve the results with an RMSE of 0.58 and RMSE of 9.15 mm.

**Table 1: The validation results based on filtering of different parameters**

| 1) coherence(%) | | | | 2) Air temperature(°C) | | | |
|---|---|---|---|---|---|---|---|
| filter out range | RMSE(mm) | R | Total number | filter out range | RMSE(mm) | R | Total number |
| <0.4 | 9.36 | 0.57 | 149 | ≥-5 | 9.61 | 0.54 | 147 |
| <0.5 | 8.6 | 0.60 | 141 | ≥-10 | 9.49 | 0.56 | 138 |
| <0.6 | 8.67 | 0.58 | 135 | ≥-15 | 9.38 | 0.66 | 74 |
| <0.7 | 8.61 | 0.6 | 127 | **≥-20** | **8.14** | **0.75** | **26** |
| 3) Elevations(m) | | | | 4) Slope(°) | | | |
| filter out range | RMSE(mm) | R | Total number | filter out range | RMSE(mm) | R | Total number |
| ≥2000 | 7.59 | **0.58** | 120 | ≥0 | 9.16 | 0.58 | 101 |
| ≥1500 | 7.58 | 0.56 | 106 | ≤0 | 10.25 | 0.54 | 51 |
| ≥1000 | 7.59 | 0.31 | 60 | remain range | | | |
| ≥600 | 6.81 | 0.39 | 9 | -5<slope<5 | 7.66 | 0.44 | 66 |
| | | | | -3<slope<3 | 8.4 | 0.4 | 53 |
| | | | | -1<slope<1 | 9.44 | -0.07 | 30 |
| 5) Snow depth(cm) | | | | 6) Snow density(kg/m3') | | | |
| remain range | RMSE(mm) | R | Total number | remain range | RMSE(mm) | R | Total number |
| 10<SD<80 | 7.84 | 0.70 | 77 | 150<density<200 | 9.15 | 0.58 | 122 |
| 20<SD<80 | 8.45 | 0.71 | 45 | 180<density<200 | 11.49 | 0.36 | 9 |
| 20<SD<40 | 8.82 | 0.63 | 25 | | | | |
| 40<SD<60 | 8.24 | 0.77 | 15 | | | | |
| 0<SD<20 | **6.96** | 0.57 | 55 | Original validation | | | |
| 0<SD<40 | 7.59 | 0.60 | 80 | | RMSE(mm) | R | Total number |
| 0<SD<60 | 7.69 | 0.66 | 95 | without filtering | 9.54 | 0.56 | 152 |
| 0<SD<80 | 7.67 | 0.68 | 100 | | | | |
| 0<SD<100 | 8.26 | 0.65 | 103 | | | | |
| 0<SD<120 | 8.64 | 0.62 | 105 | | | | |
| 0<SD<140 | 8.60 | 0.68 | 106 | | | | |

In conclusion, better validation results can be obtained by filtering temperature to below −20 °C, snow depth to 0-20 cm, and snow density to 150-200 kg/m³ in this study. However, the coherence, elevation, and slope limits do not significantly improve the inversion results. These findings are likely influenced by the distribution of properties. To describe the effect of uneven data distribution, histograms for each attribute are plotted (Fig. 21). More than half of the coherence values are in the range above 0.9, 37% of snow depths are in the 0-20 cm range, snow density is concentrated between 150-160 kg/m³, and 74% of temperature values located between −20 °C and −10 °C. Elevation and slope angles are less continuous due to the limited station distribution. These characteristics may suggest that similar properties are required to achieve inversion results comparable to ours.

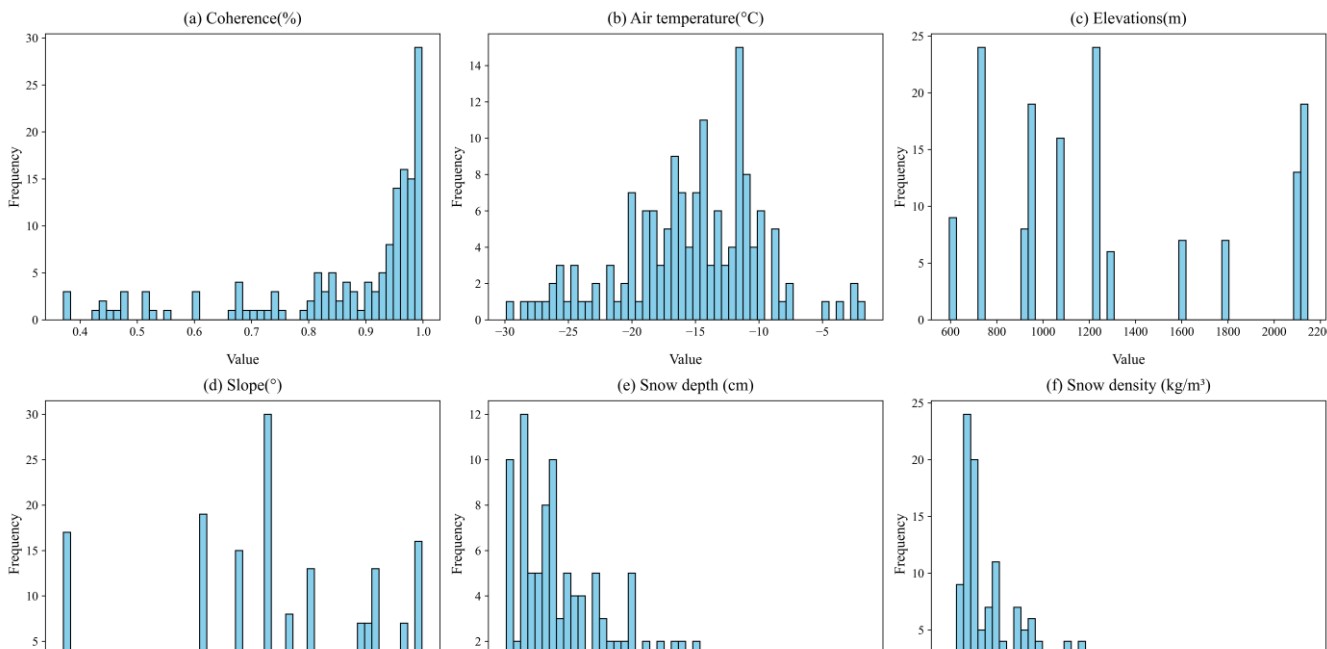

**Figure 21: Histograms of various attributes of the validation point**

## 5.2 Delta and cumulative in-situ SWE and retrieved SWE comparison at individual stations

Based on the distribution of in-situ vs. retrieved $\Delta SWE$ (rather than cumulative SWE validation) at individual stations, comparison data are grouped into two categories: one with better agreement (more clustered along the 1:1 line, Fig. 22) and one with poorer agreement (more scattered, Fig. 23).

The accuracy of cumulative SWE is directly influenced by the accuracy of $\Delta SWE$, since cumulative SWE is calculated by accumulating $\Delta SWE$ values. For example, when $\Delta SWE$ is close to the 1:1 line, most of the cumulative points is also observed

to be close to the 1:1 line, as shown in Fig. 22. However, as cumulative SWE increases, it tends to scatter more from the 1:1 line, showing a consistent trend of overestimation or underestimation through error propagation based on the time series accumulation. In contrast, there are cases where the $\Delta SWE$'s validation does not show a good relationship, yet the cumulative SWE does, as shown in Figs. 23. Similarly, with higher cumulative SWE values, data points increasingly deviate from the 1:1 line. Factors such as removing tropospheric error during processing may contribute to these discrepancies. Additionally, the

reason may be variations in the station environment and errors in the in-situ data observations. The cumulative SWE is more prone to random error, which propagates to other pairs.

Furthermore, the scattered point distribution for different years at the same station exhibits similarities. This consistency suggests that patterns of overestimation and underestimation in delta and cumulative values may stem from the station's properties or observation biases.

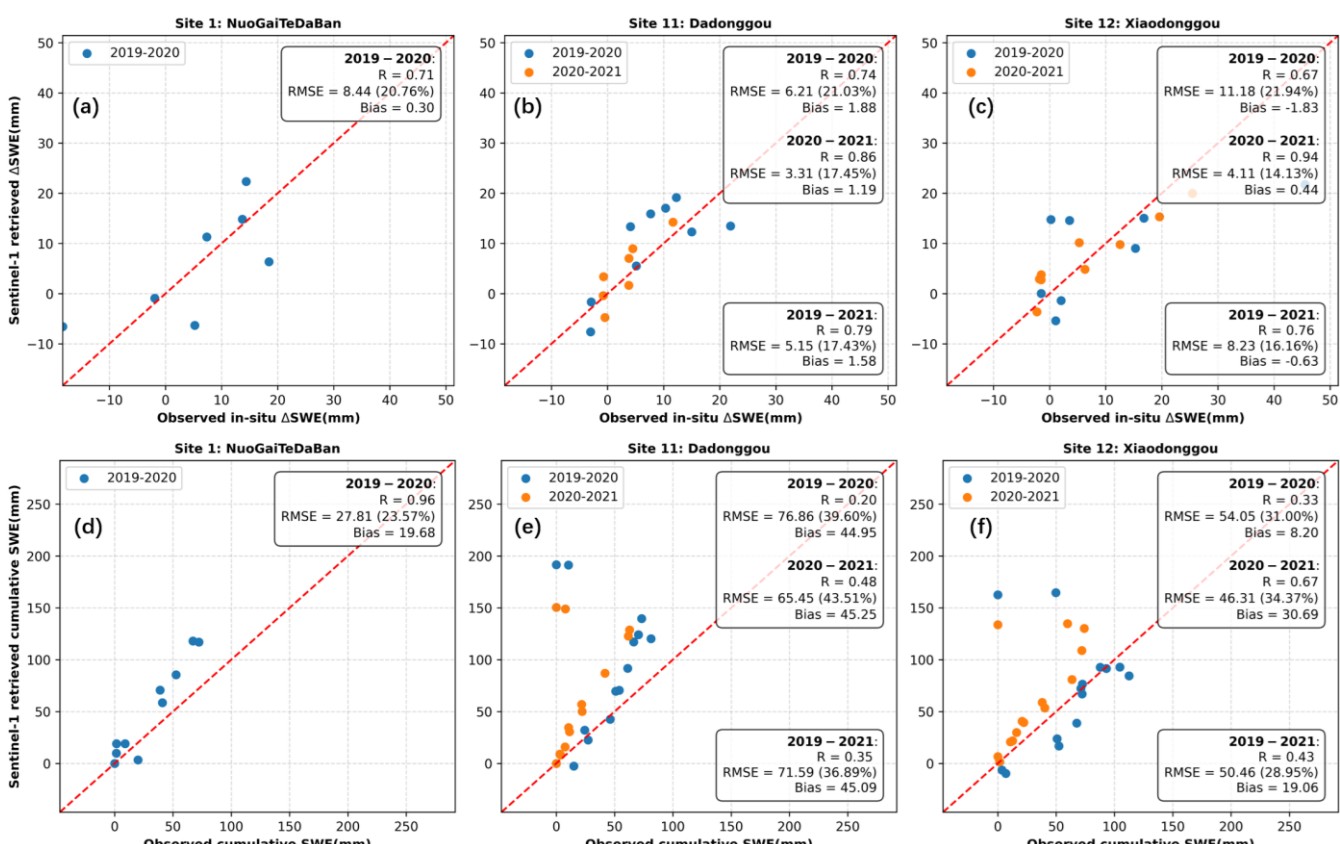

**Figure 22: Comparison between the in-situ SWE changes and the Sentinel-1 InSAR retrieved SWE changes at three stations in Altay. (The top row is $\Delta SWE$, and the bottom row is cumulative SWE; each column corresponds to the same station)**

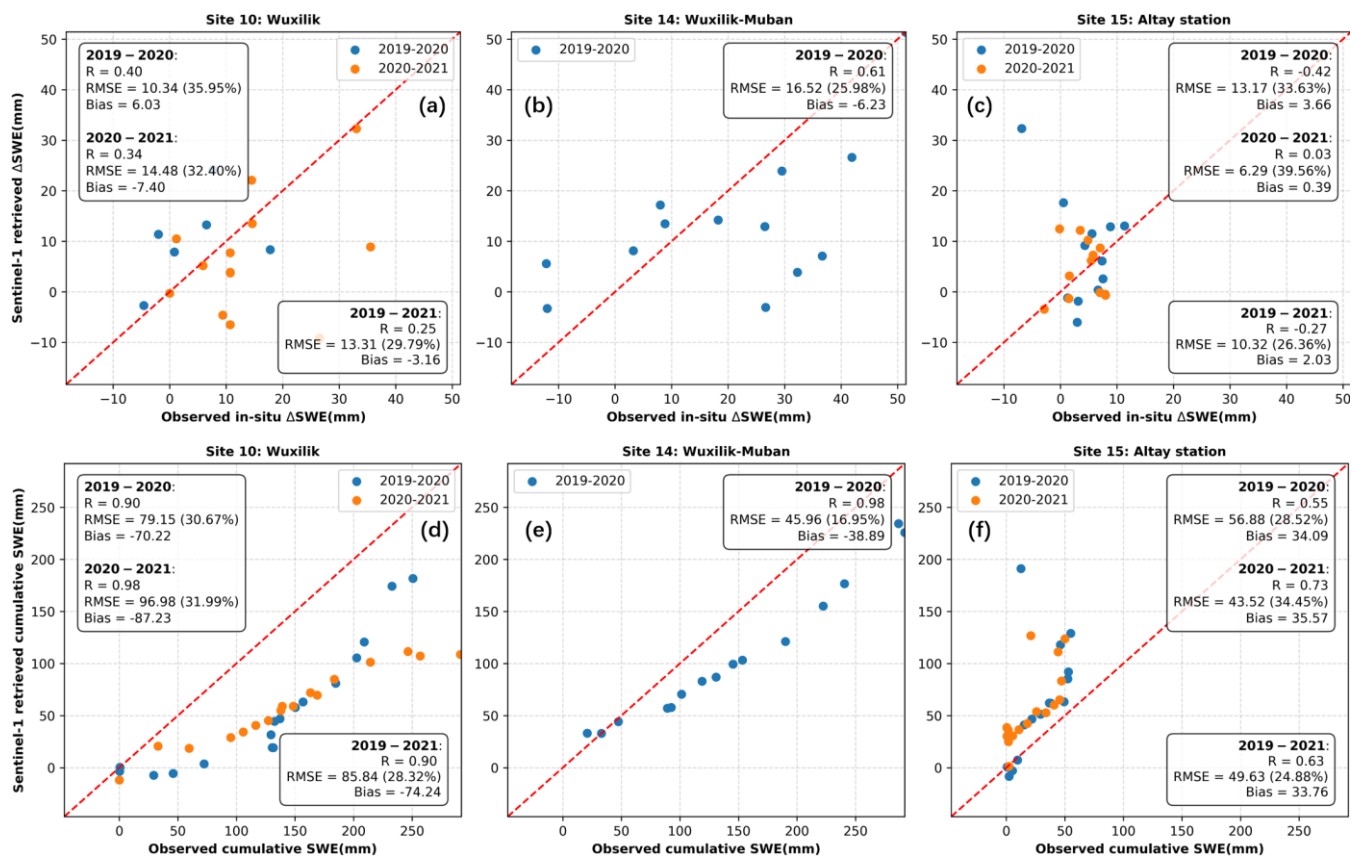

Figure 23: Comparison between the in-situ SWE changes and the Sentinel-1 InSAR retrieved SWE changes at the other three stations in Altay

## 5.3 The effects of partial phase calibration on validation of retrieved $\Delta SWE$

Based on the method described in Sect. 3.4.3, the effects of partial calibration on the validation of retrieved $\Delta SWE$ are tested. The results are shown in Fig. 24. In case (A), where no calibration is used, the validation shows a poor agreement, with no significant correlation (R = 0.14). In case (B), applying only the integer multiple of $2\pi$ part, the validation improves substantially with an RMSE of 11.9 mm (R = 0.43). In case (C), using the full calibration parameter, further improvement is observed with an RMSE of 9.5 mm (R = 0.56).

These results demonstrate that our phase calibration is essential for improving the accuracy of InSAR-based $\Delta SWE$ retrieval. While the integer multiple of $2\pi$ accounts for the main portion of the phase error, the residual phase (that is caused by data processing errors, DEM residual error, atmospheric delays, systematic phase calibration error, etc) still has a noticeable effect. Comparison to case (A), case (B) and (C) show lower RMSE and bias, as well as a higher correlation, confirming the importance of calibrating the residual phase component. It can also be observed that the overall performance is improved

through phase calibration, while some points with initially good agreement deviate from their previous alignment. In conclusion, the best accuracy can be achieved when the full calibration parameter (i.e., case (C)) is applied.

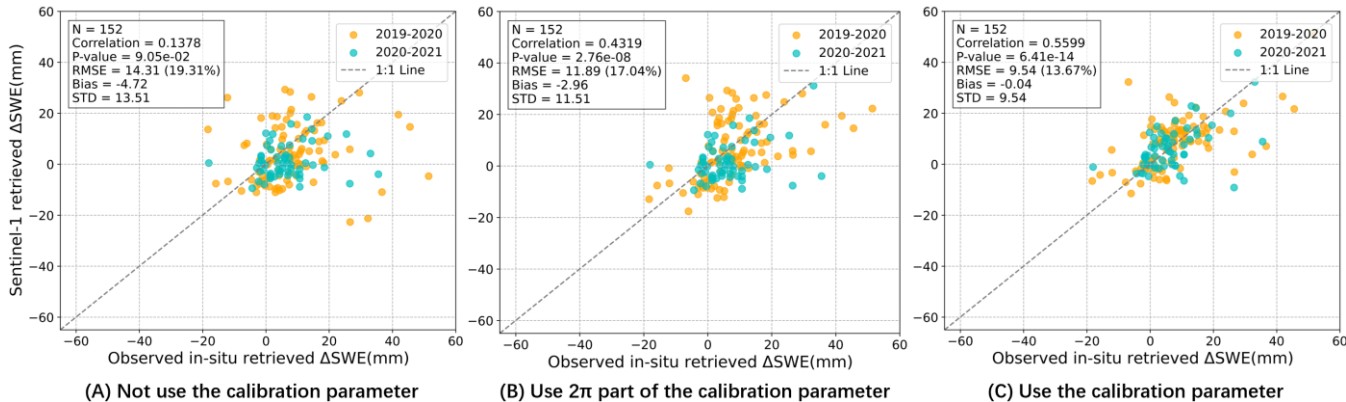

**Figure 24: Comparison of the validation results for in-situ $\Delta SWE$ in 12 days under different calibration strategies: (A) no calibration, (B) calibration using only the integer multiple of $2\pi$ part, and (C) using the full calibration parameter.**

## 6 Conclusions

In this paper, Sentinel-1 data collected every 12 days from 2019 to 2021 are used to retrieve changes in SWE (ΔSWE) and cumulative SWE throughout the entire snow season. A specific frame is selected to include 15 in-situ stations over Altay. An adequate correlation (R = 0.56) is observed between the retrieved 12-day ΔSWE and the in-situ values, with an RMSE of 9.5 mm over two years, noting that inversion results are filtered out of the wet snow. Considering that the nearly global consistent coverage offered by Sentinel-1's 12-day repeat-pass imagery, the SWE inversion using Sentinel-1 and the InSAR method presented in this study, along with the analysis of multiple factors (such as coherence and air temperature) impact on the accuracy of this retrieval technique, can be applied to other snow-covered regions.

After excluding wet snow, the retrieved cumulative SWE shows reasonable performance, with an RMSE of 40.9 mm (R=0.65). Further improvement is achieved by excluding high-elevation stations affected by early-season heavy snowfall that cause phase unwrapping errors, reducing the RMSE to 28.3 mm and increasing R to 0.70. The observations and inversion of time series cumulative SWE show consistency at several stations, albeit some stations indicate overestimations or underestimations. The scene-wide coherence, unwrapped phase, and cumulative SWE are displayed in the snow season from 2019-2021. The similarities of snow changes in two years can be found in these displays.

Moreover, a novel coherence-weighted least squares phase calibration method is introduced and validated by varying the total number of in-situ $\Delta SWE$ stations for calibration. The results show that selecting at least half of the available $\Delta SWE$ values for calibration can yield reliable InSAR-derived $\Delta SWE$ estimates. Additionally, although applying only the integer

multiple of $2\pi$ improves the results, better accuracy is achieved when the full calibration parameter is used. This suggests that the residual phase component has a pronounced contribution to the overall error and should not be ignored. Besides the results mentioned above, the factors that affect the performance of this approach are discussed, such as coherence, air temperature, and snow density. Higher coherence, lower temperatures, and more accurate snow density measurements are essential for achieving effective inversion results. Moreover, greater snowfall at higher elevations may contribute to reduced coherence.

Regarding potential limitations, on one hand, it is noted that for the InSAR method to invert SWE effectively, longer wavelengths and shorter revisit times (which improve coherence) are necessary, as well as longer time series observations for better atmospheric effect estimation. This study uses C-band data with a 12-day revisit period, which can be improved using lower frequency bands (L-band) and shorter revisit intervals. On the other hand, stations that directly measure SWE are preferred, as many stations require snow density data, introducing some uncertainty into observations. Visual interpretation errors in snow depth measurements through snow poles monitored with time-lapse cameras may also happen, particularly in sloped locations, which could amplify uncertainties. Despite these limitations, our validation results are still reasonable, providing a valuable reference for the broader application of 12-day revisited Sentinel data in SWE inversion studies.

## Data availability

The Sentinel-1 data are freely available from the European Space Agency and can be accessed via the Alaska SAR Facility (ASF, https://search.asf.alaska.edu/).

## Author contribution

Conceptualisation: JZ, YL. Writing – original draft: JZ. Writing – review & editing: YL, all authors. Data curation: JP, CX. Methodology: JZ, YL, WL, CL, ZY. Supervision: JS. Investigation: JP, CX, WM.

## Competing interests

The authors declare that they have no conflict of interest.

## Acknowledgments

The authors would like to acknowledge the European Space Agency (ESA) for providing Sentinel-1 data and the Altay Meteorological Institute for supplying ground-based observational data used in this study. ChatGPT was used to enhance the clarity of some sentences. All revisions were carefully reviewed and revised by the authors. This work was financially supported by the Ministry of Science and Technology through the National Key R&D Program of China under grant number 2022YFB3903300 and grant number 2022YFB3903301.

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
