# Peer review of "Snow Water Equivalent Retrieval and Analysis Over Altay Using 12-Day Repeat-Pass Sentinel-1 Interferometry"

_EGUsphere, 2025_

## Author Comment (AC1)

Dear Editor,

We sincerely thank the editor and the reviewers for their valuable comments and constructive suggestions, which have greatly improved our manuscript. We have carefully revised the manuscript according to the comments, and we believe the changes have addressed the concerns raised.

The major revisions are summarized as follows:

1. **Phase Calibration Method (Section 3.4)**
   - This section has been rewritten. We provide a detailed comparison of our calibration method with previous studies.
   - The phase calibration method has been updated from a least squares approach to a coherence-weighted least squares method.
   - The derivation of equations is presented in three subsequent steps, making the method clearer and more accessible.
     - A. a single InSAR scene
     - B. multiple non-redundant InSAR scenes
     - C. redundant InSAR scenes

2. **Exclusion of Wet Snow in Results**
   - The criterion for excluding wet snow is explained in Section 3.1.
   - After exclusion, all results (Figures 8-24) have been updated accordingly. Almost all performance metrics have improved compared to the results without wet snow removal. For example, the validation of $\Delta$SWE for 2019-2021 changed from R = 0.48, RMSE = 15.5 mm before exclusion to R = 0.56, RMSE = 9.54 mm after excluding wet snow.

3. **New Figure for SWE–Coherence–Topography Analysis**
   - In Section 5.1, we have added a new figure (Figure 20) to better illustrate and analyze the relationship among coherence, topography, and in-situ $\Delta$SWE measurements.

4. **Additional References**
   - We have enriched the literature review by adding relevant references on coherence, phase ambiguities, calibration methods, and SWE spatial variability.

We believe these revisions have significantly strengthened the manuscript and improved its clarity and scientific value.

Thank you again for your kind consideration. We hope the revised version now meets the requirements of the editor and reviewers.

Sincerely,
Jingtian Zhou
On behalf of all authors

**Authors' Response to Reviews of**

EGUSPHERE-2025-2329 | Research article

**Snow Water Equivalent Retrieval and Analysis Over Altay Using 12-Day Repeat-Pass Sentinel-1 Interferometry**

Jingtian Zhou, Yang Lei, Jinmei Pan, Cunren Liang, ZhangYunjun , Weiliang Li, Chuan Xiong, Jiancheng Shi, and Wei Ma

*The Cryosphere*
* * *
**RC: Reviewers' Comments**,  AR: Authors' Response,  Manuscript Text

**Reply to the first reviewer:**

**This study addresses an important gap in the application of InSAR for snow water equivalent (SWE) retrieval by leveraging Sentinel-1 C-band SAR pairs and in situ measurements collected in Altay, Xinjiang. While the theoretical foundations of InSAR for SWE estimation are well established, validation remains limited. This paper contributes valuable insight by providing a matchup dataset and a processing prototype, offering practical guidance and highlighting key limitations and uncertainties for future InSAR-based SWE applications, including those involving upcoming satellite missions.**

AR: We thank the reviewer for the positive evaluation of our work.

**However, I have several comments and suggestions that may improve the clarity and robustness of the study:**

**RC: 1. Calibration Strategy: It appears that the InSAR retrieval algorithm requires prior in situ information for calibration. Could the authors clarify whether calibration points were randomly selected, and whether the remaining data were used for validation?**

AR: Thank you very much for this insightful question.

Calibration is performed for each InSAR pair using in-situ SWE from selected stations. In order to investigate a sufficient number of in-situ stations, for each pair, a subset of available stations is randomly chosen for calibration, and the remaining stations are used for validation. For example, if we have up to 15 stations, 8 are randomly selected for calibration and the remaining 7 for validation. This random selection is repeated 100 times to assess performance when we choose 8 stations. This design allows us to evaluate how the number of calibrations influences the retrieval performance. As shown in Fig. 13 of the revised manuscript, increasing the number of calibration points consistently improves the robustness of the retrieval results, although the accuracy seems to converge after using half of the stations. Therefore, for the generation of the actual inversion product, we use all available stations to ensure high reliability and accuracy.

To avoid ambiguity, we have revised the text in Section 3.4.2 (line 332) of the manuscript to clearly state that calibration points are chosen randomly and that the remaining stations are exclusively used for validation. The revised text is shown below:

In Sect. 3.4.1, the calibration parameter for each interferometric pair is estimated by coherence-weighted least squares, where all available in situ stations are used. In this section, to validate the phase calibration method and

to assess its performance under different conditions, we test the calibration using different numbers of in-situ stations.

For each interferometer pair, only a subset of available in-situ ΔSWE observations is used to derive the calibration parameter (calibration points), while the remaining stations are reserved for validation only. Because the number of available stations is relatively small after removing missing observations or filtering wet snow (in our case, the maximum is 12), we do not apply additional selection criteria (e.g., air temperature, elevation). In particular, both the lowland (valley/plain) stations and the high-elevation mountain stations are randomly selected for calibration and validation, without any preference. Instead, a Monte Carlo random selection strategy is adopted: from all available stations, a specified number (1, 2, ⋯, up to 9) is randomly chosen for calibration, and the remaining stations (at least three) are used for validation. This random selection is repeated 100 times for each case, and the statistics of these realizations are used to quantify the performance. This procedure ensures a fair evaluation of how the number of calibration stations affects the robustness of the phase calibration and subsequent ΔSWE retrieval.

We note that, except for this specific analysis (Sect. 3.4.2) and its corresponding validation results (Sect. 4.3.2), all other retrieval results presented in this study are based on calibration using all available stations to ensure high reliability and accuracy.

**RC: 1. Calibration Strategy: I suggest an alternative approach—using data from one year for calibration and another year for validation. If the algorithm performs well under this split-sample approach, it would indicate temporal stability in the model parameters, which would be valuable for operational applications.**

AR: Thank you very much for this constructive and valuable suggestion.

We agree that testing temporal stability through a split-sample approach would be highly meaningful for operational applications. However, this approach is not feasible in our study. The estimation of calibration parameters requires both in-situ SWE observations and interferometric phases corresponding to the same acquisition period. Since both of the in-situ SWE observations and the interferometric phase vary over time (each interferogram has its unique calibration parameter), the calibration parameters estimated in one year cannot be directly transferred to another year.

**RC: 2. Assumption of Dry Snow Conditions: The theoretical model underlying the ΔSWE–Δφ relationship assumes dry snow, where the dominant reflective interface is between the snow and the ground. This assumption does not hold under wet snow conditions, where signal penetration is strongly affected. Therefore, I recommend filtering out data corresponding to wet snow conditions during the validation process. If the authors wish to explore the wet snow regime, the InSAR pair selection should be constrained to periods between two dry snow events, and then for wet snow, two acquisition from wet snow period can be selected for comparison.**

AR: We thank the reviewer for this valuable suggestion. In response, we have updated all of the validation results by filtering out data corresponding to wet snow conditions (except Figure 15 indicated by the light grey dots in Fig. 15 and the cumulative SWE curves in Figs. 16-17). Specifically, we excluded data points that meet at least one of the following criteria:

(a) Air temperature above 0°C at the station,
(b) After February 1st, when the coherence drop between two consecutive interferograms exceeded 0.3, the second interferogram and all subsequent ones were removed,
(c) Points with coherence below 0.35.

To illustrate this filtering process, Figure R1 shows the changes in air temperature, coherence, and cumulative SWE observations at the Altay station during 2019–2020. According to the above criteria, only the SWE observations within the rectangular boxes in the figure were retained for validation.

After filtering out wet snow conditions, the validation results of the ΔSWE retrieval show consistent improvements compared with those obtained without filtering in the previous submission. The revised results are now presented in Figures 10- 24 of the manuscript. In addition, we have updated the manuscript in Section 3.1 (last paragraph, line 196) to include the wet snow identification criteria, making the wet snow filtering approach explicit in the main text.

[Figure]

**Figure R1**. Air temperature, coherence, and cumulative SWE observations at the Altay station during 2019–2020

**RC: 3. Snow Wetness Screening: Although direct in situ measurements of snow wetness may not be available, backscatter signatures (e.g., sudden increase of σ⁰ ) can provide useful indicators. At a minimum, the authors could stratify the analysis by using a temporal threshold—for example, separating SAR pairs acquired before and after April—to distinguish predominantly dry versus wet snow conditions. This stratification would help reduce confounding effects and enhance the interpretability of the results.**

AR: We sincerely thank the reviewer for this insightful suggestion. Following your advice, we attempted to distinguish between dry and wet snow conditions using backscatter signatures. Specifically, we first plotted the time series of backscatter at all stations from 2019 to 2021. The Sentinel-1 VV backscatter from each station was processed by applying radiometric slope correction, spatial mean filtering with a 3 × 3 window, and temporal mean filtering with a window size of 3.

In theory, the radar response to wet snow depends on the dominant scattering mechanism: when volume scattering dominates, the increase in liquid water content (LWC) usually leads to a decrease in backscatter, but as the LWC increases further, surface scattering gradually becomes dominant, causing the backscatter to increase (Shi and Dozier, 1995; Magagi and Bernier, 2003; Long and Ulaby, 2015; Marin et al., 2022).

In principle, the onset of wet snow could be detected by identifying pivot points in the backscatter time series. However, in practice for our study area and period (2019–2021), the backscatter signals were highly unstable and did not exhibit clear or consistent pivot points (see Figures 2 for one representative stations). This made it difficult to reliably determine the transition from dry to wet snow solely from σ⁰.

Therefore, instead of relying only on backscatter signatures, we finally adopted the filtering approach described in our response to the previous comment, where wet snow conditions were excluded if any of the following criteria were met:

(a)     air temperature above 0°C at the station,

(b)     after February 1st, when the coherence drop between two consecutive interferograms exceeded 0.3, the second interferogram and all subsequent ones were removed,

(c)    points with coherence below 0.35.

This combined strategy provided a clear identification of wet snow periods (primarily start in Februray to March), which in turn improved the validation of ΔSWE retrieval.

[Figure]

**Figure R2**. Temporal evolution of the coefficient of backscattering acquired at the Wuzhiquan station in Altay during 2019–2020

**References:**

Shi, J. and Dozier, J.: Inferring snow wetness using C band data from SIR-C's polarimetric synthetic aperture radar, IEEE T. Geosci. Remote, 33, 905–914, https://doi.org/10.1109/36.406676, 1995.

Magagi, R. and Bernier, M.: Optimal conditions for wet snow detection using RADARSAT SAR data, Remote Sens. Environ., 84, 221–233, https://doi.org/10.1016/S0034-4257(02)00104-9, 2003.

Long, D. and Ulaby, F. T.: Microwave Radar And Radiometric Remote Sensing, University of Michigan Press, Ann Arbor, 1116 pp., 2015.

Marin, C., Bertoldi, G., Premier, V., Callegari, M., Brida, C., Hürkamp, K., Tschiersch, J., Zebisch, M., and Notarnicola, C.: Use of Sentinel-1 radar observations to evaluate snowmelt dynamics in alpine regions, The Cryosphere, 14, 935-956, 2020.

**Overall, this paper presents a promising step toward operational SWE retrieval using InSAR, but addressing the calibration methodology and the influence of wet snow conditions would strengthen the findings significantly.**

**Reply to the second reviewer:**

**The manuscript presents a two-year retrieval of SWE using 12-day repeat pass InSAR from Sentinel-1 C-band over the Altay region in China. The method is sound, and the results are interesting for the InSAR SWE community. However, I consider that there are several things that require attention before being accepted.**

AR: We thank the reviewer for the positive assessment of our study and for the constructive comments, which will help us to further improve the manuscript.

**General Comments:**

**RC: The manuscript could improve presentation and better explanation of the ideas. Also, I think you should consider adding more references on how your results compare to previous work and how they contribute to them. Starting from Section 3.4.1 I could only find two references for the rest of the manuscript. Some of the topics you comment (coherence, lost phase cycles/ambiguities, calibration…) have been discussed in the literature.**

AR: We appreciate the reviewer's constructive suggestion regarding the improvement of presentation and the inclusion of additional references to better position our work in the context of previous studies. Following the reviewer's advice, we have revised the manuscript accordingly:

On coherence and lost phase cycles/ambiguities: We have added several new references (e.g., Zebker and Villasenor, 1992; Jung et al., 2016; Lee et al., 2013; Engen et al., 2003; Rott et al., 2003; Ruiz et al., 2022) to highlight the impact of temporal decorrelation, phase unwrapping, and phase ambiguities on interferometric measurements (lines 191 and 664). These references make our findings in line with previous discussions on temporal decorrelation and phase wrapping in InSAR-based snow retrieval.

On calibration (Section 3.4): We revised this section to provide a clearer comparison between our approach and previous studies. Specifically, we elaborated on two commonly used strategies in the literature—using corner reflectors (e.g., Nagler et al., 2022; Dagurov et al., 2020) and in situ SWE averages (e.g., Conde et al., 2019; Oveisgharan et al., 2024). Among these, Conde et al. (2019) is a new addition. In addition, we have rewritten the text to more explicitly highlight how our method differs: instead of directly calibrating retrieved ΔSWE, we use all available in situ SWE measurements to calibrate the interferometric phase itself, ensuring a more accurate calibration. For the validation result of ΔSWE, we add a comparison to the previous work in line 467.

On SWE spatial variability (Section 4.2): We have also added references (Eppler et al., 2021; Georg et al., 2007; Deeb et al., 2011) to support our discussion of factors influencing SWE spatial variations and dynamics, such as snowfall patterns, topography, vegetation, and meteorological conditions.

We believe these additions not only improve the clarity of the manuscript but also strengthen the connection of our results to the existing literature.

**I think the phase calibration method is correct, however:**

**RC: phase calibration method: 1.   As it is now explained is a bit confusing. My suggestion is describing first the method for a single InSAR image, then extend to multiple images (redundant images as you say). Up to equation 13, is there a benefit in calibrating all images in the same step, with respect to calibrating them one by one?**

AR: We feel sorry to create such confusion and sincerely thank the reviewer for this helpful suggestion.

Regarding the first point, we have revised the description of the phase calibration method according to your advice. Specifically, we clearly introduced three new subsections and explained the method in a manner that starting from the simple single InSAR scene case and then extending it to multiple cases (see Section 3.4.1, line 269):

> In the following, we present the application of our phase calibration method in three representative cases: (A) a single InSAR scene, (B) multiple non-redundant InSAR scenes, and (C) redundant InSAR scenes (simple example).

The reviewer is correct on the second point. Up to Eq. (13) there is no difference between calibrating each interferogram individually and applying the matrix least-squares solution in calibrating all images in the same step. Indeed, Eq. (13) shows that when no redundant interferograms are involved, the solution simply reduces to an average, which is exactly equivalent to calibrating each interferogram one by one. In this case, $C_N$ is independent of $C_1, C_2, \ldots, C_{N-1}$, so solving them jointly does not provide any additional benefit.

The real benefit of the matrix formulation arises when redundant interferograms are present. In this situation, the calibration of each interferogram is no longer independent, and a joint solution is necessary. This is precisely why we adopt the matrix form, as illustrated in case (C) "redundant InSAR scenes (simple example)" in Section 3.4.1.

To make this point clearer, we have now emphasized explicitly right after the new Eq. (11) in the revised manuscript (which corresponds to the old Eq. (13) in the previous version, line 302) that the least-squares solution is equivalent to calibrating each interferogram individually in the non-redundant case, and that the advantage of the matrix formulation only becomes apparent when redundant interferograms are included.

**RC: phase calibration method: 2.** **I think one natural consideration here is weighting based on coherence. Consider adding a term to the method where observed int.phases from high coherence pixels have more weight than low coherence pixels. This way you could decrease the uncertainty derived from the noise on those pixels.**

AR: We thank the reviewer very much for this valuable suggestion. We fully agree that incorporating coherence-based weighting is an effective strategy to enhance the robustness of the estimation and to further supress the uncertainty.

Phases derived from high-coherence pixels certainly exhibit lower noise and higher reliability, and therefore should be assigned greater weight in the phase calibration estimation process. We have adopted this suggestion in the revised manuscript by introducing a diagonal weighting matrix W into the matrix formulation of Weighted Least Squares problem in Section 3.4 (e.g., Eq. 6, 8, 9, 10 and 11), where the coherence value of each station determines the weights. e.g.:

> …To estimate the phase calibration constant $C$, we employ a coherence-weighted least squares approach.
>
> Then the optimal weighted least squares coeficients for this single InAR scene is
>
> $$\hat{C} = (A^T W A)^{-1} A^T W (y - \Delta\phi) = \frac{\sum_{i=1}^{m} \gamma_i (\Delta\phi_i - y_i)}{\sum_{i=1}^{m_n} \gamma_i} \tag{6}$$
>
> where $W = diag(\gamma_1, \gamma_2, \ldots, \gamma_m)$ is an $m \times m$ diagonal weight matrix based on coherence values $\gamma_i$.
>
> where $\gamma_i$ represents the coherence value at the $i$-th measurement location. This formulation assigns a

larger weight to phases derived from locations with higher coherence, enhancing the robustness of the calibration…

**RC: phase calibration method: 3.     Specify clearly that y is calculated from the ground data. What is the point of Eq (6)? Same for Eq (15)).????**

AR: We thank the reviewer for raising these points for the phase calibration method. We have revised the manuscript to explicitly state that "$y$ represents the calculated interferometric phase from in-situ $\Delta SWE$ measurements. " at line 268.

Regarding the previous Eq. (6) and (15): in the earlier version of the manuscript, Eq (6) defines the total number of equations (or observations) $m$ when we have N number of integerograms. To distinguished from this non-redundant case, we use $M$ in the old Eq. (15) for the redundant example.

$$m = \sum_{i=1}^{N} m_i \tag{6}$$

$$M = m_{12} + m_{23} + m_{13} \tag{15}$$

In response to the reviewer's comment, to avoid confusion, we have revised the manuscript so that these two equations are no longer presented as stand-alone equations. Instead, they are embedded in the text and explained sequentially in the following sections: Section (B) Multiple Non-redundant InSAR Scenes ( line 283), and Section (C) Redundant InSAR Scenes: A Simple Example (line 311).

**RC: The results from the coherence are interesting but limited. Adding discussion on topography or coherence against in-situ dSWE could broaden the discussion.**

AR: We sincerely thank the reviewer for this valuable suggestion. Following the recommendation, we have performed additional analysis to investigate the relationships between coherence, topography, and in-situ ΔSWE. The corresponding analysis has been incorporated into the main text, and the results are presented in the newly added Fig. 20 in Section 5.1, where multiple influencing factors on ΔSWE retrieval are systematically discussed.

[Figure]

**Figure 20: Analysis of coherence and topographic effects on in-situ ΔSWE. (A) Coherence vs. in-situ Δ SWE, (B) Elevation vs. in-situ ΔSWE, (C) Coherence vs. Elevation.**

- ➤ **Coherence** vs **in-situ ΔSWE (Fig. 20A):** A weak negative correlation is observed ($R = -0.32$). Most points cluster at high coherence values (>0.8) and within a relatively narrow ΔSWE range of $-10$ mm to 20 mm. The weak correlation may be because the majority of ΔSWE values fall in this small interval while coherence remains high. A few points with ΔSWE > 30 mm show low coherence (<0.6), leading to the phase wrapping problem, since the phase-to-ΔSWE conversion (Eq. (3)) indicates that one $2\pi$ cycle corresponds to ~30 mm of ΔSWE, beyond which phase unwrapping errors can occur.

- ➤ **Elevation** vs **in-situ ΔSWE (Fig. 20B):** A weak positive correlation is found ($R = 0.29$). This likely reflects the tendency for greater SWE changes at higher elevations. This implies that higher elevations generally experience more snowfall and accumulate thicker snowpacks, leading to greater SWE changes.

- ➤ **Coherence** vs **Elevation (Fig. 20C):** A negative correlation is detected ($R = -0.34$). At higher elevations, coherence values exhibit a wider distribution, including both high and low values, whereas lower elevations tend to cluster at higher coherence levels. This pattern, also visible in Fig. 19(1) and (3), may be associated with greater snowfall at higher elevations, which could lead to reduced coherence.

Incorporating these results has allowed us to expand the discussion and offer a more nuanced analysis, as encouraged by the reviewer.

**RC: Some plots are way too packed with information, making them very hard to read. E.g., fig 22.**

AR: We sincerely apologize for the lack of clarity in the original figure. To improve readability, we have revised the previous Figure 22 and Figure 21, which are now updated as Figures 23 and 22 in the manuscript.

[Figure]

**Figure 22:** Comparison between the in-situ SWE changes and the Sentinel-1 InSAR retrieved SWE changes at three stations in Altay. (The top row is ΔSWE, and the bottom row is cumulative SWE; each column corresponds to the same station)

[Figure]

**Figure 23:** Comparison between the in-situ SWE changes and the Sentinel-1 InSAR retrieved SWE changes at the other three stations in Altay

**Specific Comments:**

RC: In Section 2.2.1, maybe comment at what time is the flight pass already here.

AR: We thank the reviewer for the comment. The text in Section 2.2.1 has been revised to include the satellite overpass time. The revised sentence now reads at line 115: "All data correspond to a descending flight direction with an overpass time of approximately 00:13 UTC the local Beijing Time being 08:13 (UTC+8) , path 19, frame 434."

RC: Lines 147 to 150: do you mean that the spatial resolution from ERA-5 is too coarse? It is not clear for me what's the meaning of these sentences.

AR: We feel sorry to these unclear sentences. Yes, we mean that the spatial resolution of ERA5 is relatively coarse compared to that of Sentinel-1 InSAR data.

Specifically, ERA5-Land hourly data provides snow density data at a spatial resolution of 0.1° × 0.1° (~9 km), while Sentinel-1 InSAR observations used in this study have a spatial resolution on the order of tens of meters. As a result, the snow density values derived from ERA5 represent spatial conditions over large areas and may not capture the fine-scale spatial variability of snow density at the scales of InSAR pixels. This scale mismatch

could introduce errors in the validation when using ERA5 snow density values to calculate SWE observations for the snow depth stations.

We have revised the relevant sentence in the manuscript to clarify this point at lines 147 to 151.

**RC: Line 165: This is correct, but the sentence is hard to understand as it is now. Consider splitting it.**

AR: We appreciate this comment and agree that the original sentence was overly complex. The sentence has been split and revised for clarity:

> **Original sentence:** The InSAR SWE retrieval algorithm considers that the signal penetration through the snow layer to the ground and the main contribution of backscattering at the ground covered by dry snow is coming from the snow-ground interface, and the volume scattering effect on the interferometric can be neglectable confirmed by the ground-based experiment (Matzler, 1996).

> **Revised sentence**: The InSAR SWE retrieval algorithm considers the signal penetration through the snow layer to the ground. The primary backscattering contribution from dry snow-covered terrain originates from the snow–ground interface, while the volume scattering effect on the interferometric phase is negligible, as confirmed by ground-based experiments (Mätzler, 1996).

**RC: Line 188: two things here, what do you mean that dPhi is estimated from the unwrapped InSAR phase? I guess you use Eq. 3 for retrieve dSWE?**

AR: Thank you for these important questions. Regarding your first point: the term "dPhi" indeed refers to the unwrapped interferometric phase. The previous wording was not precise and has therefore been removed in the revision.

As for the second question, you are correct: Eq. (3) is used to retrieve dSWE from dPhi, as it represents the inversion relationship between dSWE and the unwrapped phase.

**RC: Line 189 to 192: I'll argue that the main limitation, in particular for C-band and 12 days temporal baseline is decorrelation, and vegetation if there is any on the study area. If you are going to comment on phase unwrapping (lost phase cycles) you could indicate the amount of mm of dSWE that are equivalent to dPhi=$2\pi$.**

AR: Thank you for raising these critical points. We agree with the reviewer that temporal decorrelation, especially given the 12-day temporal baseline and C-band wavelength used in this study, is indeed a primary limitation for InSAR-based SWE retrieval, particularly in vegetated areas. Vegetation can cause decorrelation and introduce volume scattering effects that are not accounted for in the dphi-to-dSWE model. Specifically, a phase shift of dPhi=$2\pi$ corresponds to a dSWE of approximately 30 mm for C-band, as already noted in later Line 334 of Section 4.1 in previous manuscript. This clarification has been added to the limitations part to provide a concrete reference for readers.

The sentence in the limitation paragraph has been modified as follows:

> **Original:** "The main advantage of this method is its simplicity and does not need prior knowledge, while the main limitation is the problem of the phase unwrapping when the SWE change is larger than typically 1-2 wavelengths. Still, the wet snow absorption during the snow melt and when large snowfalls occur will limit the ability of this method (Storvold et al., 2006). This method is designed for dry snow conditions."

**Revised:** "The main advantage of this method is its simplicity and a reduced reliance on a priori information. However, its application is constrained by several factors: (1) temporal decorrelation (Zebker and Villasenor, 1992; Jung et al., 2016; Lee et al., 2013), which is particularly critical for C-band data with a 12-day revisit cycle and can be severe over vegetated terrain; (2) the phase unwrapping problem (Engen et al., 2003; Rott et al., 2003; Leinss et al., 2015), which occurs when the SWE change is about larger 30mm in our cases ($2\pi$ phase change corresponds to 30mm SWE changes); and (3) signal attenuation during snowmelt or in the presence of wet snow, which limits the method to dry snow conditions (Storvold et al., 2006)."

**RC: Figure 3: If you want to refer to distances CA, DE, etc. I think these should be explained in the text (?). Also theta_s appears only in the drawing. Better simplify the figure.**

AR: Thanks for these helpful suggestions regarding Figure 3. We have revised the figure as follows and the accompanying text accordingly.

As recommended, the labels for distances CA, DE, etc., which were previously only mentioned in the figure caption, have now been explicitly defined and explained in the main text (line 178) to improve clarity.

Regarding $\theta_s$, we agree that it appears only in the figure. However, we have retained it because it clearly illustrates the critical refraction process at the air-snow interface (to distinguish from incidence angle $\theta$), which is important to the geometric derivation of the signal path.

To simplify the figure as suggested, we have removed the redundant incidence angle $\theta$ at the bottom of the figure, as another incidence angle indicator remains clearly visible in the upper part.

[Figure]

**Figure 3: Propagation path of radar wave through atmosphere with snow-free and snow-covered ground for a fixed pixel. $\theta$ is the incidence angle, $\theta_s$ is the refracted angle in snow, $\varepsilon_s$ is the real part of the permittivity of the snow, $\varepsilon_{air}$ is the real part of the permittivity of the air, and $\Delta Z_s$ is the increase in snow depth between the snow-free and snow-covered ground images.**

**RC: Line 218: Phase calibration is explained in Section 3.4!**

AR: We sincerely apologize for this oversight. The sentence has been corrected to Sect. 3.4 in the revised manuscript.

**RC: Lines 253 to 255: not sure what is the point about the local incidence angle here. Both sentences seem to contradict each other.**

AR: We thank the reviewer for pointing out this confusing statement. The term "local incidence angle" was incorrectly used and has been removed.

Regarding the contradiction, we intended to distinguish between two different calibration strategies. previous studies (e.g., Oveisgharan et al., 2023) calibrated retrieved ΔSWE using the average of all in-situ ΔSWE. In contrast, we calibrate the interferometric phase. If local incidence angles are taken into account, a uniform phase offset across the scene results in different ΔSWE offsets at each site. This makes a scene-wide ΔSWE calibration invalid, whereas calibrating the phase directly is more appropriate. The text has been revised accordingly in 257.

**RC: Line 295: can you explain what "Monte Carlo random selection" is, and how it is applied?**

AR: Thank you for your comment and for raising this question.

To clarify what we mean by "Monte Carlo random selection": In our context, we adopt a random selection strategy instead of selecting calibration points based on their specific attributes (such as airtemperature, elevation, etc.), which is not practical in our case due to the limited number of available stations (a maximum of 12) and their narrow attribute range. We randomly select a subset of stations for calibration and use the remaining stations for validation. This process is repeated multiple times (100 realizations), following the Monte Carlo principle. This repetition ensures that, for a given number of stations, the calibration results are not dependent on any specific random selection of stations and allows us to obtain representative statistical outcomes.

For better clarity, we have also provided additional explanations in Section 3.4.2 of the revised manuscript.

**RC: Section 4.1: 1.   It is true that phase ambiguities can occur but based on the calibration methodology this may not be a problem in practice. Can authors comment on it? 2.    Can authors comment on why same interferograms show a very similar decorrelation pattern from (assuming these are georeferenced) north-west to south-east? Is it related to topography?**

AR: 1. Our calibration constant is indeed designed to estimate and remove the scene-wide phase bias of the interferogram, which may include integer multiples of $2\pi$. Since the calibration relies on a limited number of reference sites, it may not fully capture spatially varying phase ambiguities. However, local unwrapping errors mainly depend on coherence and may persist even after calibration. In particular, areas with low coherence cannot be reliably unwrapped. To mitigate these issues, our workflow includes unwrapping correction, multilooking, and filtering which are applied before calibration. This improves phase quality and minimizes the practical impact of phase ambiguities. We have added the above clarification to the Section 4.1 (line 390).

2. We appreciate the reviewer's observation. Yes, the decorrelation pattern is indeed related to topography. As indicated in the elevation map ( red rectangle area in Figure 1, Figure R3(A), and newly added Figure 20 as the reviewer suggested), the north-east region is characterized by relatively high elevations (above ~2000 m), while the south-west part of the study area is lower (around ~1000 m) and relatively flat. The observed north-west to south-east decorrelation coincides with the higher-elevation regions, where the complex topography likely contribute to reduced coherence. This topographic dependence explains why the decorrelation pattern is consistently observed across interferograms. We have added the above explanation to the end of Section 4.1.

[Figure]

**Figure R3:** Topography and coherence in Altay

**RC: Section 4.2: 1.  I suggest you introduce Equation (22) already here as Figs 8 and 9 are showing that. 2.  The titles of subfigures in 8 and 9 should be a single date, from the secondary.**

AR: 1.We sincerely thank the reviewer for this constructive suggestion. We agree that Equation (22) should be introduced earlier, as Figs. 8 and 9 already refer to the cumulative SWE based on this equation. We have revised the manuscript accordingly to improve clarity and consistency:

**4.2 Spatiotemporal distribution/variations in cumulative SWE**

All of the time series retrieved $\Delta$SWE are used to calculate the retrieved cumulative SWE at each date from the start date of our satellite's observation date, which can be expressed as

$$SWE(t_{i+1}) = SWE(t_0) + \sum_{t_j=t_0}^{t_i} \Delta SWE(t_j, t_{j+1}) \qquad (14)$$

where $t_0$ is the start date (September 5, 2019 or September 11, 2020), $SWE(t_{i+1})$ is the cumulative (or absolute) SWE on the date of $t_{i+1}$, and $\Delta SWE(t_j, t_{j+1}) = SWE(t_{j+1}) - SWE(t_j)$. For example, the cumulative SWE at 20190929 (yyyymmdd) is the summation of the cumulative SWE at the initial date of 20190905, $\Delta SWE$ (20190905,20190917), and $\Delta SWE$ (20190917,20190929).

2.We appreciate the reviewer's careful remark. Following the suggestion, we have revised the titles of subfigures in Figs. 8 and 9 to display only a single date corresponding to the secondary acquisition.

[Figure]

**Figure 8: Spatiotemporal distribution of cumulative SWE in Altay during the 2019–2020 snow season. (Both this figure and the following one show the SWE variation relative to the first reference scene, with a 12-day cumulation interval. The SWE of the reference scene is set to 0. The reference scene for this figure is September 5, 2019. The cumulation's end dates are shown at the top of each sub-figure. The red rectangles mark areas A, B, and C to describe the SWE variations across different regions. To improve comparison, these are colored in the same range.)**

[Figure]

Figure 9: Spatiotemporal distribution of cumulative SWE in Altay during the 2020–2021 snow season. (The reference scene for this figure is September 5, 2020.)

**RC: Line 340 to 343: This is not really obvious on the plots.**

AR: Thank you for this valuable comment. We agree that our original explanation of the coherence pattern was too detailed and not fully supported by strong evidence. In the revised manuscript, we have simplified this description in line 416: ...Moreover, the coherence pattern may reveal human activities (e.g. human grazing activities in September).

**RC: Section 4.3.1: Carefully consider when is dSWE and when is SWE. Are the points used for calibration excluded for Fig. 10? In my opinion you should.**

AR: We sincerely thank the reviewer for these important comments.

(1) Regarding the first suggestion, we have carefully checked the use of dSWE and SWE in Section 4.3.1, and the incorrect usages have been corrected in the revised manuscript.
(2) We understand the reviewer's concern. In Section 4.3.2, we provide an independent analysis by splitting the dataset into calibration and validation subsets, showing that using only half of the points for calibration is already sufficient, although including more points can further improve the robustness. Nevertheless, for the actual product inversion, we employed all available points, which is also consistent with the previous studies (e.g., Oveisgharan et al., 2023). In Fig. 10, all in-situ SWE data were used, but only to estimate a scene-wide phase constant rather than to locally fit each point. Thus, including more points improves the robustness of this constant, ensuring high reliability and accuracy. Corresponding clarifications have been made in the revised manuscript (lines 253, 341 and 510):
  a) Line 254: In our case, it is difficult to identify a fixed phase reference point because snow accumulation and ablation affect the entire scene, and corner reflector deployment is labor-intensive. Therefore, we follow the second strategy and use all available in situ stations for calibration rather than a subset. Using a larger number of spatially distributed stations reduces random errors and minimizes potential biases

introduced by individual stations. This approach ensures a more stable and unbiased calibration of the InSAR-derived ΔSWE.
b)  Line 341: Instead, a Monte Carlo random selection strategy is adopted: from all available stations, a specified number (1, 2, ···, up to 9) is randomly chosen for calibration, and the remaining stations (at least three) are used for validation. This random selection is repeated 100 times for each case, and the statistics of these realizations are used to quantify the performance. This procedure ensures a fair evaluation of how the number of calibration stations affects the robustness of the phase calibration and subsequent ΔSWE retrieval.
c)  Line510: Therefore, all other retrieval results presented in this study are based on calibration using all available stations to ensure high reliability and accuracy.

**RC: Section 4.4: I think the sign for the complex dielectric permitivitty is incorrect. Can you comment on the decrease on SWE in the late season? How can you calculate it since assuming wet snow the retrieval is not possible?**

AR: We sincerely thank the reviewer for those comments.

(1) **On the sign for the complex dielectric permittivity:**

The sign issue refers to the ± in the complex dielectric permittivity, in which convention it is defined. In other words, either one can be used as long as it is defined clearly. Our sign convention, with a minus sign in the imaginary part, follows Ulaby et al. and Hallikainen et al.

**References:**

Long, David, and Fawwaz Ulaby. Microwave radar and radiometric remote sensing. Artech, 2015.

Hallikainen, M., Ulaby, F., and Abdelrazik, M.: Dielectric properties of snow in the 3 to 37 GHz range, IEEE transactions on Antennas and Propagation, 34, 1329-1340, 1986.

(2) **For the decrease on SWE in the late season**

Concerning the reviewer's second question on the decrease of SWE in the late season, a decrease of SWE is indeed observed because this period corresponds to the melting season when air temperature rises. For the retrieval results, although the dry-snow algorithm is no longer valid in wet-snow conditions, the accumulated SWE is calculated based on the full time series. Therefore, the decrease is still reflected in the accumulated SWE curve. Moreover, for very low wetness in the early stage of the melting process, the impact of snow melting on the SWE decrease is still captured. This clarification has been added in Section 3.2 (line 582) of the revised manuscript.

(3) **As for the retrieval under wet-snow conditions**

In this section, we applied the dry-snow algorithm to the entire dataset, including both dry- and wet-snow periods, to maintain continuity in the cumulated SWE time series. We note that the failure of the inversion algorithm is gradual, and during the late-season wet-snow period, the retrieval shows only a limited dynamic range, indicating that the algorithm is not suitable under such conditions. To clarify this, we explained the behavior from the perspective of the reduced penetration capability of microwaves in wet snow.

In the updated manuscript, we clarify that for ΔSWE, wet-snow periods were explicitly excluded, and only dry-snow conditions were retained. The exclusion criteria are described in the last paragraph of Section 3.1 (line 196). However, since Section 4.4 focuses on cumulative SWE, which is obtained by integrating the dSWE time series. Therefore, some wet-snow points were retained in the cumulation process (as indicated by the light grey dots in

Fig. 15 and the cumulative SWE curves in Figs. 16-17). Nevertheless, these wet-snow points were excluded in the final cumulative SWE evaluation here. We have clarified this point in the revised manuscript in line 542.

**Grammar:**

**RC: The manuscript could use a revision of English and format (justifications). Remove duplicated definition of acronyms. I have write down some catches, but there are more:**

AR: We thank the reviewer for carefully checking the English and formatting issues. Following the suggestions, we have revised the manuscript thoroughly to improve clarity and readability, removed duplicated definitions of acronyms, and corrected the grammatical and formatting errors noted by the reviewer. The specific points raised have been addressed as follows:

**RC: Page 2, Line 53: This sentence is hard to understand, consider rewriting.**

The sentence has been rewritten for clarity.

> **Original sentences:** However, a single parameter retrieval of SWE is challenging because radar backscatter is a function of several other parameters, including snow density, snow depth, snowpack liquid water content, snow stratigraphy, snow grain size, and soil/vegetation conditions, as well as systematic factors (frequency, polarisation).

> **Revised sentences:** However, a single parameter retrieval of SWE is challenging. This is because radar backscatter depends on multiple factors, such as snow density, snow depth, liquid water content, stratigraphy, grain size, and soil/vegetation conditions, as well as systematic factors (frequency and polarisation).

**RC: Page 2, Line 59: No need for point in parenthesis.**

The redundant point in the parenthesis has been removed.

> **Original sentences:** …(depending on wavelength, e.g., 15 mm at L-band, 3.75 mm at C-band.) by …

> **Revised sentences:**… (e.g., 15 mm at L-band, 3.75 mm at C-band.) by …

**RC: Page 2, Line 60: Check reference from Guneriussen is duplicated.**

The duplicated reference to Guneriussen has been corrected.

**RC: Page 3, Line 67: was explored.**

The verb tense has been corrected in the revised manuscript.

**RC: Page 3, Line 73: was…**

The wording has been corrected accordingly.

**RC: Line 105: by the?**

The grammar issue has been fixed.

**RC: Line 163: I don't know if algorithm is the correct word here.**

We agree and have replaced "algorithm" with "method".

**RC: Line 256: as follows**

Revised to "as follows."

**RC: Line 261: why this equitation has a different form than EQ (3)?**

Eq. (3) expresses the theoretical relationship between phase change ($\Delta\phi$) and SWE ($\Delta SWE$):

$$\Delta\phi = 2k_i \cdot \frac{\alpha}{2}\left(1.59 + \theta^{\frac{5}{2}}\right) \cdot \Delta SWE$$

In Section 3.4.1, we reformulate this expression by defining y as a substitution term, i.e.,

$$\boldsymbol{y} = 2k_i \cdot \frac{\alpha}{2}\left(1.59 + \theta^{\frac{5}{2}}\right) \cdot \Delta SWE.$$

There are two reasons for using this different form. First, the substitution avoids confusion with later equations, particularly Eq. (5), where the symbol $\boldsymbol{y}$ on the left-hand side denotes the phase change derived from ground-based measurements, while the $\Delta\phi$ on the right-hand side refers to the interferometric phase observed by the satellite. Defining $\boldsymbol{y}$ in advance helps ensure that the two different $\Delta\phi$ terms are not conflated. Second, this form also enables a more concise, flexible, and matrix-friendly representation when extending the phase calibration to multiple interferometric pairs.